# Self-regulated reversal deformation and locomotion of structurally homogenous hydrogels subjected to constant light illumination

Kexin Guo[1,2,3,4], Xuehan Yang[1,4], Chao Zhou[2,3] & Chuang Li [1]✉

Environmentally adaptive hydrogels that are capable of reconfiguration in response to external stimuli have shown great potential toward bioinspired actuation and soft robotics. Previous efforts have focused mainly on either the sophisticated design of heterogeneously structured hydrogels or the complex manipulation of external stimuli, and achieving self-regulated reversal shape deformation in homogenous hydrogels under a constant stimulus has been challenging. Here, we report the molecular design of structurally homogenous hydrogels containing simultaneously two spiropyrans that exhibit self-regulated transient deformation reversal when subjected to constant illumination. The deformation reversal mechanism originates from the molecular sequential descending-ascending charge variation of two coexisting spiropyrans upon irradiation, resulting in a macroscale volumetric contraction-expansion of the hydrogels. Hydrogel film actuators were developed to display complex temporary bidirectional shape transformations and self-regulated reversal rolling under constant illumination. Our work represents an innovative strategy for programming complex shape transformations of homogeneous hydrogels using a single constant stimulus.

Shape deformation in response to environmental stimuli is ubiquitous in nature and is an essential feature that endows living organisms with actuating and adapting functions critical for their survival. Elegant examples range from the hygroscopic opening and closing of a pinecone for seed dispersal[1], the fast shuttling of the Venus flytrap for insect capture[2], to the $Ca^{2+}$ gradient-mediated contraction of animal muscles[3]. These biological morphing systems have inspired people to unearth the key paradigms and features that underlie efficient natural shape transformation to further guide the design and synthesis of artificial shape-morphing materials[4,5], including shape-memory polymers[6,7], liquid crystal elastomers (LCEs)[8], hydrogels[9], and others.

In particular, hydrogels have attracted unusual attention and offer unique advantages in constructing shape-morphing materials due to their intrinsic high-water content and hydration/dehydration-mediated reversible volume change[10,11], both of which are common actuating features shared with natural living organisms. In response to external stimuli such as ions[12,13], force[14,15], heat[16,17], light[18–20], humidity[21], magnetic field[22], and electric field[23,24], shape-morphing hydrogels can change their volume, shape, structure and/or other physiochemical properties for promising applications in artificial muscles[25–27], fatigue-resistant materials[28,29], encryption devices[30], actuators[31–33], and soft robotics[34,35]. To achieve the desired anisotropic deformation instead of

[1]Key Laboratory of Precision and Intelligent Chemistry, Department of Polymer Science and Engineering, University of Science and Technology of China, Hefei 230026, China. [2]CAS Key Laboratory of Nano-Bio Interface, Division of Nanobiomedicine and i-Lab, Suzhou Institute of Nano-Tech and Nano-Bionics, Chinese Academy of Sciences, Suzhou 215123, China. [3]School of Nano-Tech and Nano-Bionics, University of Science and Technology of China, Hefei 230026, China. [4]These authors contributed equally: Kexin Guo, Xuehan Yang. ✉e-mail: lichuang21@ustc.edu.cn

isotropic volumetric expansion or contraction, the fabrication of shape-deformable hydrogels currently relies on either a sophisticated design of heterogeneous hydrogels with layered[13,30], oriented[20], gradient[36] anisotropic structures or complex control and manipulation of nonuniform external stimuli[11,37,38]. The former preheterogenization strategy principally leads to fixed anisotropy in hydrogels, which limits their shape-changing mode and flexibility, while the latter typically requires complex external assistance for shape transformation, which lacks self-adaptivity and intelligence. In contrast, living organisms can respond to external environmental stimuli in a highly adaptable and intelligent manner by flexibly adjusting their internal compositions/structures to enable temporary and erasable structural anisotropy to drive on-demand shape deformation[4]. For example, the reversible movement of pinecone scales in response to changes in humidity is attributed to the dynamic reorientation of internal cellulose microfibrils within cellular walls[4]. This motivated scientists to develop structurally homogenous hydrogels that are capable of displaying self-regulated dynamic structural anisotropy under a specific stimulation, therefore leading to on-demand temporary or transient shape transformation[18,39,40].

Photoresponsive hydrogels[41] with high spatiotemporal controllability represent a unique type of such material that can produce a transient structural gradient for dynamic shape morphing. Due to the instantaneous delivery and time-dependent gradual penetration of light, a structural gradient along the thickness direction can be generated photochemically[25,42] or photothermally[17–19] in a short time and subsequently eliminated at prolonged illumination, typically resulting in reversible bending-flattening deformation of hydrogel films[33,42,43]. Although the dynamic shape change of such hydrogels can be reversibly controlled by external light manipulation, the current deformation mode under constant illumination is monotonous and can usually generate unidirectional bending deformation toward or away from the irradiation source. Herein, we report the molecular design of self-adaptable hydrogels that are capable of exhibiting self-regulated transient bidirectional deformation subject to constant light illumination. The deformation reversal mechanism was found to originate from self-regulated sequential descending-ascending charge variations in two mixed spiropyrans under continuous constant illumination, leading to sequential volume contraction and expansion throughout the thickness direction. The bidirectional deformation performance was highly tunable by controlling the mixing formulation of two spiropyrans, which was demonstrated in parallel in two separate molecular systems. Various transient bidirectional shape transformations and temporary bidirectional rolling locomotion under constant light illumination have been demonstrated. Our work paves the way for the development of bioinspired intelligent hydrogel materials with self-regulated nonmonotonic deformation modes subject to a continuous constant stimulus for future applications in smart actuators and soft robotics.

## Results

### Design principles of MCH(1+2) hydrogels with continuous contraction-expansion capability under constant illumination

In this work, we synthesized two polymerizable spiropyran compounds (see Supplementary Figs. 1–3 for details), which are designed to display significantly distinct photoisomerization kinetics and totally opposite charge variation capabilities upon irradiation with visible light. As shown in Fig. 1a, compound MCH1 experiences a net-charge growth upon irradiation by isomerization from protonated zwitterionic merocyanine (MCH) to negatively charged spiropyran (SP), while compound MCH2 undergoes a net-charge decrease under the same conditions due to its isomerization from positively charged MCH⁺ to the neutral SP form. Owing to the presence of the electron-donating methoxy group, MCH2 exhibited a much slower photoisomerization rate ($K_2 = 0.003$ s$^{-1}$) relative to that ($K_1 = 0.759$ s$^{-1}$) of MCH1, enabling

dramatically asynchronous isomerization behavior in response to the same irradiation (Fig. 1b and Supplementary Fig. 4). These two compounds with a mixing molar ratio of 1:1 were synchronously copolymerized in the presence of N-isopropylacrylamide (NIPAAm) monomers and N,N′-methylenebisacrylamide (MBAAm) crosslinkers to form MCH(1+2) hydrogels with two spiropyran moieties distributed randomly (see hydrogel preparation in Methods for details). According to our design illustrated in Fig. 1c, taking MCH1 and MCH2 as a whole, the initial state of the MCH(1+2) hydrogel in the dark is assigned to be positively charged, with MCH1 and MCH2 being zwitterionic and positively charged, respectively. However, upon irradiation with short-term incident light, MCH1 isomerizes to its negatively charged form, while MCH2 maintains its positively charged form owing to their asynchronous sensitivity to the same illumination. Therefore, the MCH(1+2) hydrogel experiences a transient neutral state, with MCH1 and MCH2 being negatively and positively charged, respectively. The transition from the initial positively charged state to the transient neutral state is expected to lead to a volumetric contraction of the MCH(1+2) hydrogel because a decrease in the total net charge lowers the hydrophilicity of the polymer chains, which would drive water molecules to diffuse out of the hydrogel. With prolonged irradiation, MCH2 isomerizes to its neutral SP form, resulting in a final negatively charged state of the MCH(1+2) hydrogel, with MCH1 and MCH2 being negatively charged and neutral, respectively. Volumetric expansion is expected to occur in the transition process from the transient neutral state to the final negatively charged state as an increase in the total net charge enhances the hydrophilicity of the polymer chains and drives the diffusion of water molecule into the hydrogel[42]. In this way, our designed MCH(1+2) hydrogel is able to sequentially display time-dependent bimodal volumetric contraction-expansion when subjected to constant illumination.

In our experiments, we indeed observed such a first-contracting and then-expanding behavior in the MCH(1+2) hydrogel under continuous irradiation (Fig. 1d). In contrast, the single-component control hydrogels exhibited only monotonous volume changes in volume under the same conditions (the MCH1 hydrogel expanded, and the MCH2 hydrogel contracted). By averaging the expansion value of the MCH1 hydrogel and the contraction value of the MCH2 hydrogel, we obtained a calculated volume change in the MCH(1+2) hydrogel (green curve), which was found to disagree and behave oppositely with our experimental result (blue curve), suggesting that the observed non-monotonous contraction-expansion of the MCH(1+2) hydrogel is not a simple composite of MCH1 and MCH2 but rather a result of the synergistic interaction of those two as a whole. Specifically, although the single-component MCH1 hydrogel itself displayed a volume expansion upon irradiation, the isomerization of MCH1 to SP1 in the mixed MCH(1+2) hydrogel did not lead to an expansion but instead gave an opposite volume contraction due to the simultaneous presence of positively charged MCH2 quenching/neutralizing the negative charge of SP1. Similarly, the isomerization of MCH2 to SP2 does not generate a volume contraction like a single-component MCH2 hydrogel but instead leads to a volume expansion of the mixed MCH(1+2) hydrogel due to the presence of SP1. The role that MCH1 (or MCH2) plays in the mixed MCH(1+2) hydrogel is opposite to that in the individual MCH1 (or MCH2) hydrogel, generating a calculation curve of volume change with the opposite trend, as shown in Fig. 1d. These results indicated that the nonmonotonous contraction-expansion phenomenon observed in our MCH(1+2) hydrogel is rooted in the synergistic interaction between MCH1 and MCH2, which should be considered not separately but synergistically as a whole. Therefore, hydrogel contraction or expansion upon irradiation directly depends on the total net charge decrease or increase in imbedded spiropyran[42], and this principle not only applies to single-component MCH1 and MCH2 hydrogels but also applies to two-component MCH(1+2) hydrogels, in which the total net charge of two spiropyrans plays a decisive role. In addition, we

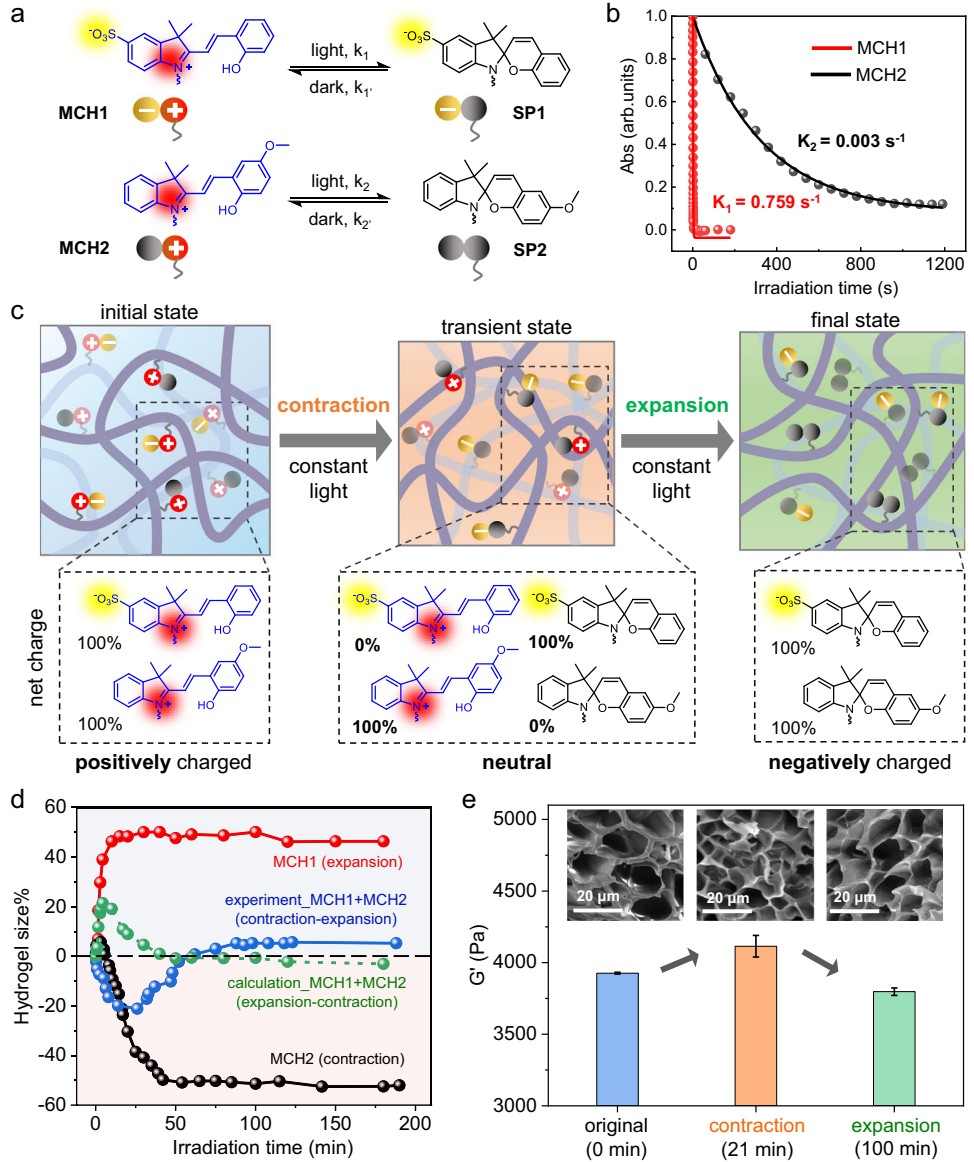

**Fig. 1 | Design of light-induced bidirectional contraction-expansion of MCH(1 + 2) hydrogels. a** Chemical structures and net charge states of the protonated ring-opening merocyanine form (MCH1, MCH2) in the dark and ring-closed spiropyran form (SP1, SP2) before and after irradiation with visible light. The photoisomerization rate $k_1$ is designed to be much greater than that of $k_2$. **b** Plot of the characteristic absorbance of MCH1 at 422.5 nm (red) and MCH2 at 450.3 nm (black) versus irradiation time (450 nm, 154.6 mW/cm²), followed by fitting to the ExpDec1 function to obtain the photoisomerization rates. **c** Schematic representation of the transient sequential contraction-expansion of MCH(1 + 2) hydrogels due to the stepwise descending-ascending variation in the total net charge. **d** Plot of hydrogel size (%, defined as the percentage change in diameter of the dish-shaped hydrogel) as a function of two-sided irradiation (8.99 mW/cm²) time. The green dashed curve represents the calculated hydrogel size, obtained by averaging the changes in size of MCH1 (red, expansion) and MCH2 (black, contraction). **e** Rheological measurements of MCH(1 + 2) hydrogels before (blue) and after irradiation (8.99 mW/cm²) for 21 min (orange) and 100 min (green). The insets are corresponding SEM images showing the changes in microporosity. Error bars represent standard deviations of data collected from three separate samples.

observed a bidirectional change in the rheological modulus (first increase and then decrease) as well as a morphological reversal in the pore size (first decrease and then increase) of MCH(1 + 2) hydrogels (Fig. 1e), which are completely different from the monotonic changes in both the modulus and porosity observed in single-component MCH1 and MCH2 hydrogels (Supplementary Fig. 5). These results verified that our MCH(1 + 2) hydrogels are indeed able to display sequential changes in volume as designed.

## Bidirectional bending of MCH(1+2) hydrogel films under constant illumination

To obtain asymmetric shape deformation behaviors instead of isotropic volume contraction and expansion, constant incident light is applied to the MCH(1 + 2) hydrogel thin film (0.5 mm thick) from one side so that the time-dependent penetration of light creates an isomerization gradient across the thickness direction. As illustrated in Fig. 2a, the MCH(1 + 2) hydrogel film displays a designed bidirectional bending deformation upon irradiation from above, sequentially creating a transient bimodal positive-negative bending deformation. Such a nonmonotonous deformation subjected to constant illumination could be attributed to the time-dependent penetration of light, which generated a reversal deformation gradient between the top and bottom areas. Although both the top and bottom areas of the hydrogel film undergo charge variation in the sequence of positive-neutral-negative upon irradiation, there will be an obvious response lag in the bottom area because the light reaches it later than in the top area.

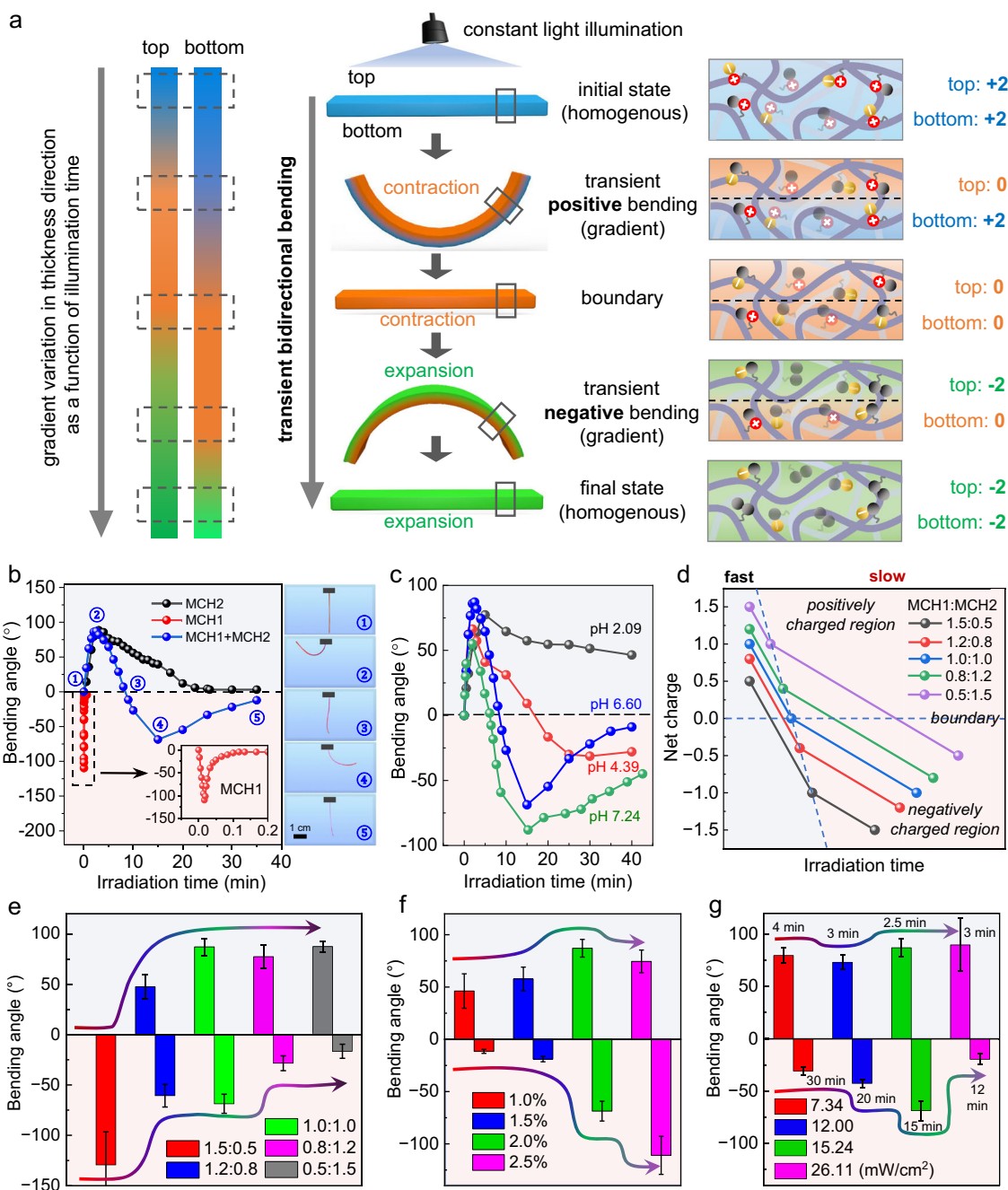

**Fig. 2 | Light-induced reversal bending deformation of MCH(1 + 2) hydrogel thin films. a** Schematic representation of transient bidirectional positive and negative bending deformation of a ribbon-shaped MCH(1 + 2) hydrogel upon irradiation from above. **b** Plot of the bending angles of the MCH1 hydrogel (red), MCH2 hydrogel (black), and MCH(1 + 2) hydrogel (blue, molar ratio is 1:1) as a function of irradiation duration (450 nm, 15.24 mW/cm²). On the right are photographs of the bending geometry of the MCH(1 + 2) hydrogel ribbon at a specific irradiation time. **c** Plot of the positive and negative bending angles of MCH(1 + 2) hydrogel ribbons upon irradiation at different pH values. **d** Plot of the net charge changes in mixed MCH1 and MCH2 with variable ratios as a function of irradiation time, showing positively and negatively charged regions with tunable charge transition time points. **e** Plot of the bending angles of MCH(1 + 2) hydrogel ribbons with variable mixing ratios under constant irradiation (450 nm, 15.24 mW/cm²). **f** Plot of the bending angles of MCH(1 + 2) hydrogel ribbons with variable total grafting densities (MCH1:MCH2 = 1:1). **g** Plot of the bending angles of MCH(1 + 2) hydrogel ribbons containing a fixed mixing ratio of 1:1 and a fixed total grafting density of 2.0% upon irradiation with variable light intensities. The labeling time represents the illumination time when the hydrogel reaches its maximum bending angle. Error bars in (**e**–**g**) represent standard deviations of data collected from three separate samples.

Therefore, charge gradient reversal occurs through the thickness under constant illumination, giving rise to sequential bidirectional bending deformation. We indeed observed such bidirectional bending behavior of the MCH(1 + 2) hydrogel strip in our experiments (Fig. 2b, Supplementary Fig. 6 and Supplementary Movie 1). In contrast, single-component hydrogels can only display monotonous bending-flattening deformation under the same irradiation conditions.

The bidirectional bending deformation was found to be related to the environmental pH, and a relatively high pH value facilitates the reversal of such deformation (Fig. 2c) due to the spontaneous deprotonation and ring closure of MCH at higher pH (Supplementary Fig. 7).

In addition to being equivalent, varying the mixing molar ratio of MCH1 and MCH2 turns out to be a flexible and straightforward strategy for tuning the performance of the MCH(1 + 2) system

(Supplementary Fig. 8). Theoretically, the net charge of MCH(1 + 2) in the initial and final states as well as its positive-negative charge reversal time is highly dependent on the mixing molar ratio (Supplementary Figs. 9 and 10). The former process, from the initial positive charge state to the neutral state under irradiation, corresponds to the contraction gradient for a positive bending deformation, while the latter process, from the neutral to negative charge state, governs the expansion gradient for a negative bending deformation. A molar ratio of 1:1 results in symmetric positive and negative bending angles under constant irradiation. However, when the ratio deviates from 1:1, a higher molar ratio of MCH1 results in a shorter positive-negative charge reversal time and a greater absolute value of negative charge change in the mixed MCH(1 + 2) hydrogel (Fig. 2d). MCH(1 + 2) hydrogels with variable mixing ratios of MCH1 and MCH2 exhibited adjustable bidirectional bending behaviors. As shown in Fig. 2e, except for the 1.5:0.5 ratio, which only exhibited negative bending, we observed reverse bending deformation for the other ratios in our study under constant illumination. It was found that a higher molar ratio of MCH1 (or MCH2) produces a larger negative (or positive) bending angle due to the greater total negative (or positive) charge change under irradiation. Only if the ratio of MCH1 was close to that of MCH2 could a symmetric bidirectional bending angle be obtained (Supplementary Fig. 11). With a fixed mixing ratio of 1:1, we next investigated the total grafting ratio of spiropyran compounds, as shown in Fig. 2f, and we found that a higher grafting ratio allows for a larger bending angle but might break the bending angle symmetry (Supplementary Fig. 12). Therefore, we fixed the total grafting ratio of spiropyran at 2 mol% for our subsequent study. In addition, the light intensity applied to the hydrogel film also plays an important role in determining the reversal of bending deformation. As shown in Fig. 2g and Supplementary Fig. 13, 15.24 mW/cm² was found to display optimum bidirectional bending behavior, giving rise to both large positive and negative bending angles with good symmetry. These results indicated that MCH(1 + 2) hydrogel films are indeed able to display the expected bidirectional bending deformation when subjected to constant illumination, and their reversal bending behaviors are highly tunable by adjusting the mixing molar ratio, total grafting ratio, environmental pH and irradiation conditions.

**Extending the bidirectional deformation mechanism to MCH(3+2) hydrogels**

To verify the universality of the bidirectional deformation reversal mechanism, we designed and synthesized two additional compounds, MCH3 and MCH4, that exhibit significantly distinct photoisomerization rates under the same irradiation conditions (see Supplementary Figs. 1 and 14–15 for details). Specifically, MCH3 is an analog of MCH1 that experiences a net-charge growth upon irradiation but with a methoxy group, while MCH4 is an analog of MCH2 that undergoes a net-charge decrease upon irradiation but lacks a methoxy group (Supplementary Fig. 16). In this way, it is guaranteed that there is always a significant difference between the photoisomerization rates of these two compounds (Supplementary Fig. 17), which leads to a sequential charge descending-ascending variation under constant irradiation (Supplementary Figs. 18–20). The prepared MCH(3 + 4) hydrogel was found to exhibit similar bidirectional bending deformation upon exposure to the same illumination conditions applied to the MCH(1 + 2) hydrogel (Supplementary Fig. 21 and Supplementary Movie 1). As a control, the prepared MCH(1 + 4) hydrogel was not found to display any bidirectional deformation in our experiments under the same conditions (Supplementary Fig. 22 and Supplementary Movie 1), which we attributed to the fact that both MCH1 and MCH4 have similar relatively fast photoisomerization rates (Supplementary Fig. 23), limiting sequential charge variation and the creation of a transient neutral state (Supplementary Figs. 24–26). These results further suggested that the mismatch of photoisomerization rates

between two spiropyran moieties plays a critical role in determining the reversal of bending deformation in mixed hydrogels.

Nevertheless, it should be noted that there are still possibilities to endow hydrogels with deformation reversal capability using a combination of spiropyrans with similar but slow photoisomerization rates. Figure 3a shows an example of the combination of MCH3 and MCH2, both of which possess relatively slow photoisomerization rates (Fig. 3b and Supplementary Fig. 27) due to the presence of methoxy groups in their chemical structure. In this case, both MCH3 and MCH2 gradually isomerize to their SP form with a similar but slow rate under the same irradiation, providing a sufficient time window for the creation of a transient state with a neutral net charge (Fig. 3c). The transformation from an initial positive charge to such a transient neutral state allows for volume contraction of the MCH(3 + 2) hydrogel. With prolonged illumination, such a temporary neutral state could eventually transfer to its final negatively charged state owing to the complete ring closure of both MCH3 and MCH2, leading to an opposite volume expansion when subjected to constant illumination. Figure 3d shows the detailed change in hydrogel volume as a function of irradiation time, which indeed exhibited a first contraction and then expansion trajectory. Not surprisingly, upon irradiation, the single-component hydrogel only exhibited a monotonic change in volume where the MCH3 hydrogel expanded while the MCH2 hydrogel contracted. The calculated change in volume obtained by averaging the changes in volume of the MCH3 and MCH2 hydrogels matches well with our experimental curve, which could be attributed to the numerical proximity of the photoisomerization rates of MCH3 and MCH2. Similarly, we used rheological measurements and SEM imaging to confirm the sequential bimodal changes in the mechanical strength and porous morphology of the MCH(3 + 2) hydrogel during the whole irradiation process (Fig. 3e), which are different from those of the monotonous changes in the single-component MCH3 and MCH2 hydrogels (Supplementary Fig. 28). These results demonstrated that, in addition to selecting a pair of spiropyrans with significantly different photoisomerization rates, such as MCH(1 + 2) or MCH(3 + 4), choosing a pair of spiropyrans with similar slow photoisomerization rates, such as MCH(3 + 2), turns out to be another useful approach to develop hydrogels with reversal volume changes subjected to constant illumination.

Similar to the MCH(1 + 2) hydrogel investigated above, we further explored the asymmetric bending deformation of the MCH(3 + 2) hydrogel thin film by irradiation with constant light from one side. Although the MCH(3 + 2) hydrogel is expected to sequentially exhibit transient bimodal positive-negative bending behavior when subjected to constant light illumination, the detailed gradient reversal across the thickness direction of this hydrogel varies from that of the MCH(1 + 2) hydrogel. As shown in Fig. 4a, light first penetrates the top area of the thin film, leading to a charge variation in the sequence of +2 → 0 → −2 due to the similar photoisomerization rates of MCH3 and MCH2. Meanwhile, the bottom area is treated as unchanged with a net charge of +2. Therefore, the top area first contracts and then expands, resulting in positive bending and flattening deformation. With prolonged irradiation, light penetrates the whole thickness, and the bottom area also undergoes a charge change in the same sequence of +2 → 0 → −2, while the top area maintains −2. This subsequently drives the thin film to experience negative bending and flattening deformation. During the whole irradiation process, the MCH(3 + 2) hydrogel underwent self-regulated bidirectional bending due to the variation in charge of the mixed spiropyrans (Fig. 4b and Supplementary Movie 1), while the single-component hydrogel only displayed monotonous bending-flattening deformation under the same irradiation conditions. The bidirectional deformation of the MCH(3 + 2) hydrogel showed a similar pH dependence as that of the MCH(1 + 2) hydrogel (Fig. 4c and Supplementary Fig. 29).

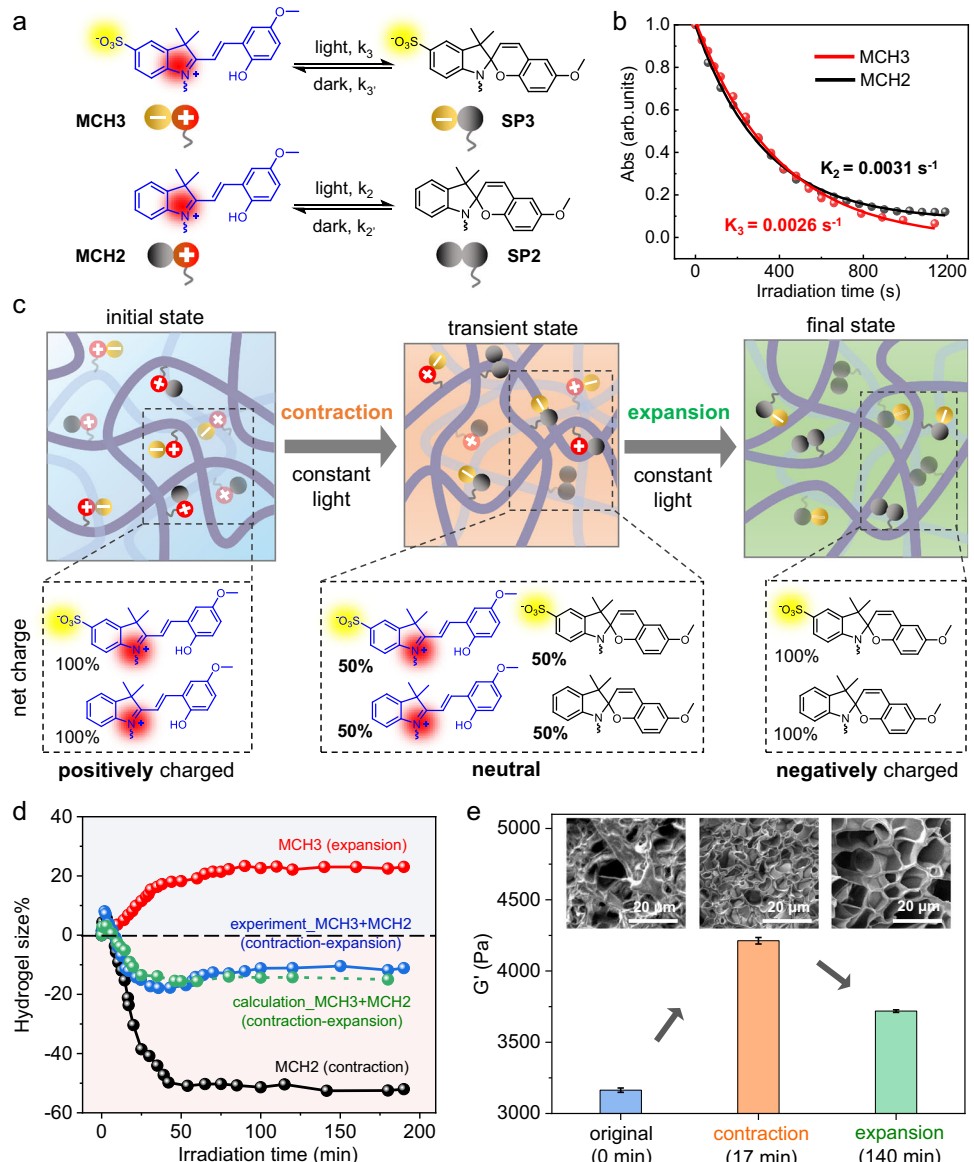

**Fig. 3 | Design of light-induced bidirectional contraction-expansion of MCH(3 + 2) hydrogels. a** Chemical structures and net charge states of the protonated ring-opening merocyanine form (MCH3, MCH2) in the dark and ring-closed spiropyran form (SP3, SP2) before and after irradiation with visible light. The photoisomerization rate $k_3$ is designed to be as slow as that of $k_2$. **b** Plot of the characteristic absorbance of MCH3 at 456 nm (red) and MCH2 at 450 nm (black) versus irradiation time (450 nm, 154.6 mW/cm²), followed by fitting to the ExpDec1 function to obtain the photoisomerization rates. **c** Schematic representation of the transient sequential contraction-expansion of MCH(3 + 2) hydrogels due to the

stepwise descending-ascending variation in the total net charge. **d** Plot of hydrogel size (%, defined as the percentage change in diameter of the dish-shaped hydrogel) as a function of two-sided irradiation (8.99 mW/cm²) time. The green dashed curve represents the calculated hydrogel size obtained by averaging the changes in size of MCH3 (red, expanded) and MCH2 (black, contracted). **e** Rheological measurements of MCH(3 + 2) hydrogels before (blue) and after irradiation (8.99 mW/cm²) for 17 min (orange) and 140 min (green). The insets are corresponding SEM images showing the changes in microporosity. Error bars represent standard deviations of data collected from three separate samples.

By changing the ratio of MCH3 and MCH2 to adjust the positive/negative charge proportions (Supplementary Figs. 30–32), we can control the charge reversal time point (Fig. 4d), the bending angle and the bending symmetry (Fig. 4e and Supplementary Fig. 33). Furthermore, varying the total spiropyran grafting ratio (Fig. 4f and Supplementary Fig. 34) and controlling the irradiation intensity (Fig. 4g and Supplementary Fig. 35) are also powerful strategies for modulating the bidirectional bending performance.

## Self-regulated biomimetic shape transformation under constant illumination

Based on the robust deformation reversal capability we developed above, the MCH(1 + 2) hydrogel was chosen as an example to further

construct a bioinspired bidirectional shape transformation subjected to constant illumination. As shown in Fig. 5a, MCH(1 + 2) hydrogels with various initial shapes, including fish tails, crosses, stars, and brushes, are held from the top and display self-regulated positive-negative bending behaviors upon irradiation with constant light from the left (Supplementary Movie 2). These transient bending configurations somewhat mimic or replicate certain features of movements and shape changes we observed in nature, generating morphologies of fish-tail-wagging, dragonfly flying, flower-blooming and mimosa-leave-closing. Importantly, such bidirectional deformation was highly reversible, and the material could be cycled multiple times by alternatively switching the light on and off (Supplementary Fig. 36), during which no obvious decrease in deformation performance was observed (Fig. 5b).

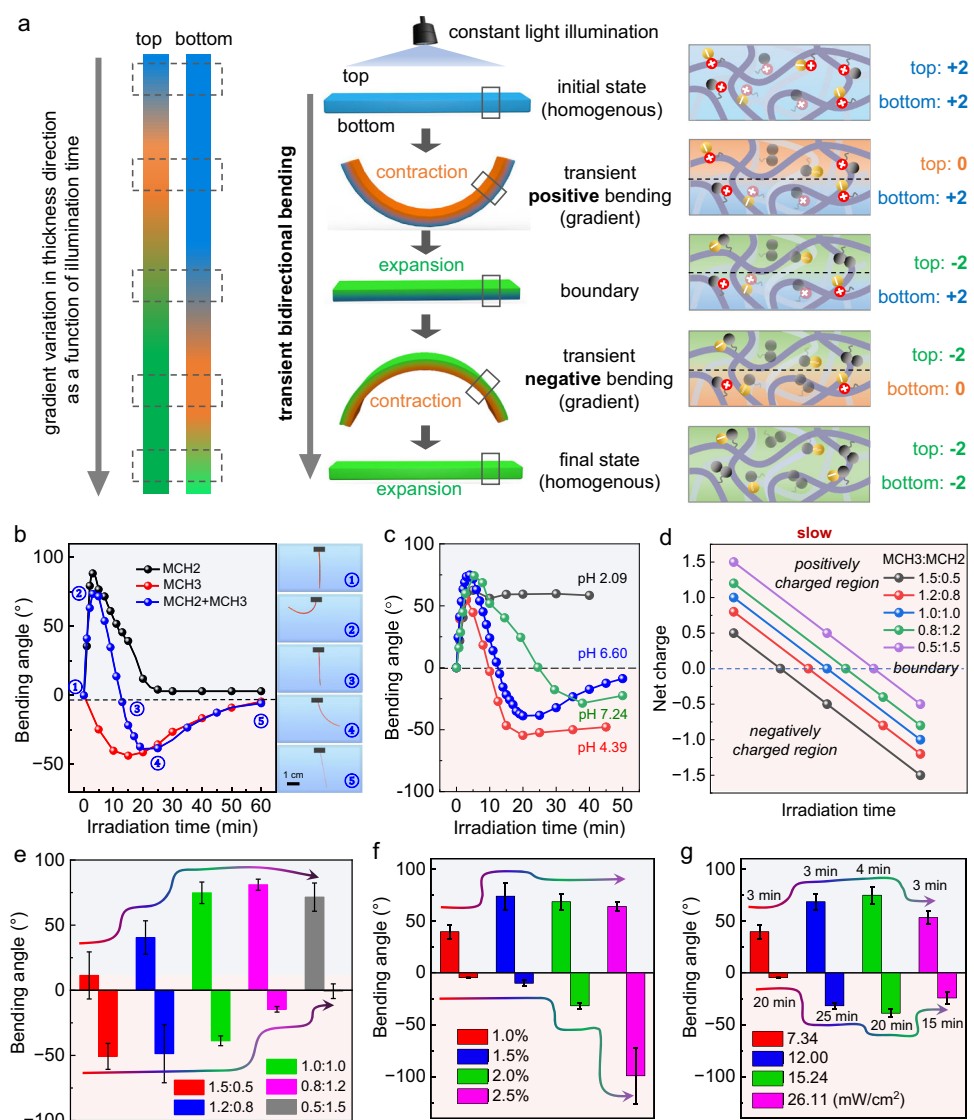

**Fig. 4 | Light-induced reversal bending deformation of MCH(3 + 2) hydrogel thin films. a** Schematic representation of transient bidirectional positive and negative bending deformation of a ribbon-shaped MCH(3 + 2) hydrogel upon irradiation from above. **b** Plot of the bending angles of the MCH3 hydrogel (red), MCH2 hydrogel (black), and MCH(3 + 2) hydrogel (blue, molar ratio is 1:1) as a function of irradiation duration (450 nm, 15.24 mW/cm²). On the right are photographs of the bending geometry of the MCH(3 + 2) hydrogel ribbon at a specific irradiation time. **c** Plot of the positive and negative bending angles of the MCH(3 + 2) hydrogel ribbons upon irradiation at different pH values. **d** Plot of the net charge changes in mixed MCH3 and MCH2 with variable ratios as a function of irradiation time, showing positively and negatively charged regions with tunable charge transition time points. **e** Plot of the bending angles of MCH(3 + 2) hydrogel ribbons with variable mixing ratios under constant irradiation (450 nm, 15.24 mW/cm²). **f** Plot of the bending angles of MCH(3 + 2) hydrogel ribbons containing variable total grafting densities (MCH3:MCH2 = 1:1). **g** Plot of the bending angles of MCH(3 + 2) hydrogel ribbons containing a fixed mixing ratio of 1:1 and a fixed total grafting density of 2.0% upon irradiation with variable light intensities. The labeling time represents the illumination time when the hydrogel reaches its maximum bending angle. Error bars in (**e**–**g**) represent standard deviations of data collected from three separate samples.

Furthermore, a dual-star-shaped hydrogel initially sitting on a flat surface was found to undergo programmable transient bidirectional bending behaviors depending on the direction of irradiation by external constant light (Fig. 5c). By utilizing the same irradiation direction from the top (pathway 1) or bottom (pathway 3) or by alternating irradiation directions (pathway 2), we obtained various programmed temporary deformation configurations in a designed sequence, such as up-up/down-down (pathway 1), down-down/up-up (pathway 3), and down-up/up-down (pathway 2). It should be noted that all these transient deformation behaviors are achieved from a single hydrogel sample, which could relax to its initial state in the dark and could be used for the next photoprogramming event, demonstrating the rich deformation modes and high reusability of our hydrogels.

## Self-regulated bidirectional rolling motion under constant illumination

To endow our hydrogels with locomotion functionality, we prepared O-ring-shaped hydrogels (see bidirectional rolling motion in Methods for details) with the aim of achieving self-regulated bidirectional rolling motion driven by constant light illumination. Figure 6a illustrates the rolling motion reversal mechanism of an O-ring-shaped MCH(1 + 2) hydrogel subjected to constant light illumination. Initially, the exposed bottom-left area of the O-ring hydrogel contracts first due to the creation of a gradient throughout the thickness direction, leading to a decrease in its curvature and subsequent a shift in its center of gravity to the left. A torque is therefore generated and accumulated strong enough to drive the O-ring roll toward the light source on a glass substrate. However, as

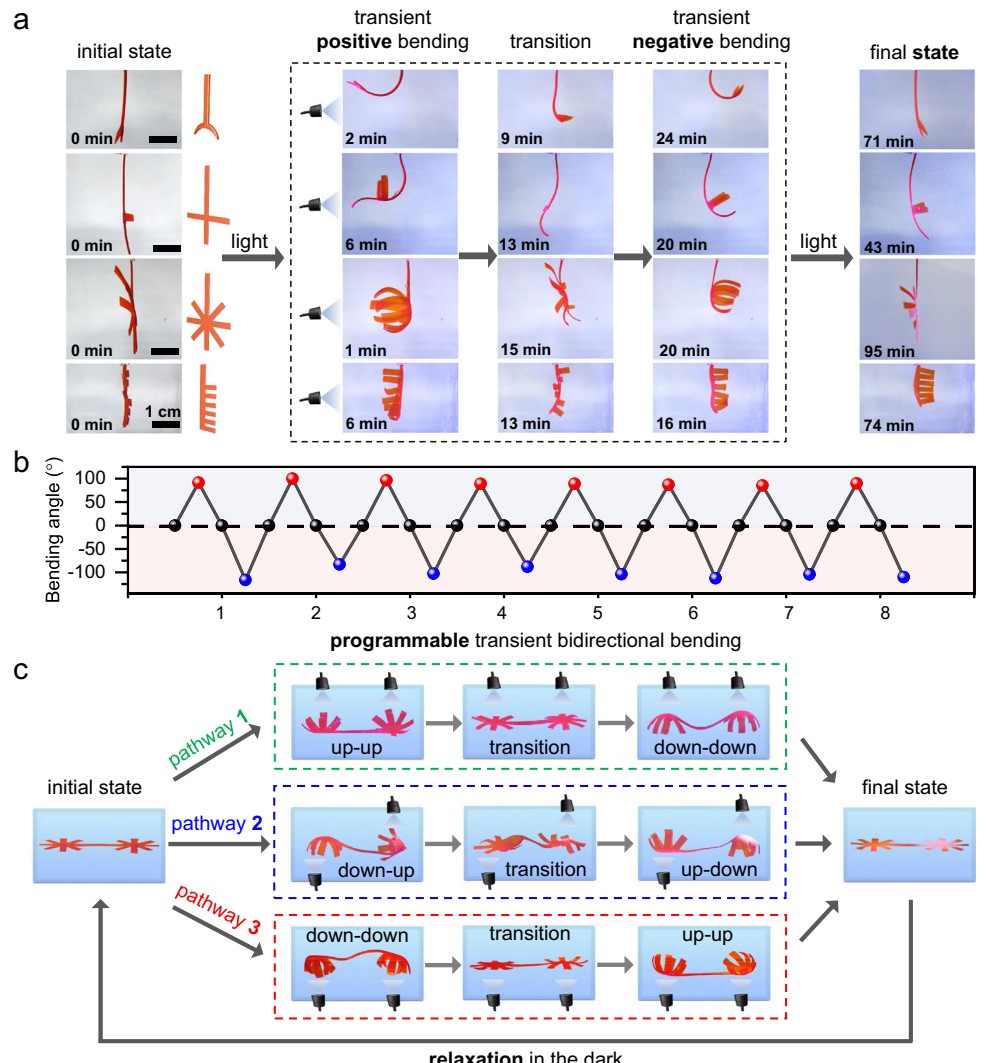

**Fig. 5 | Self-regulated bidirectional shape transformation. a** Photographs of transient bidirectional bending deformation of MCH(1 + 2) hydrogels with multiple inverted shapes, including fish tails, crosses, stars, and brushes. **b** Plot of transient positive (red) and negative (blue) bending angles of a star-shaped MCH(1 + 2) hydrogel for eight repeated cycles by alternatively switching light (450 nm, 15.24 mW/cm²) on and off. For each cycle, the hydrogel was incubated in acidic water (5 mM) for 8 h to fully relax to its original state. **c** Photographs of a single dual-star-shaped MCH(1 + 2) hydrogel that displays programmable transient deformation configurations by manipulation of different irradiation pathways.

it approaches the light source with prolonged illumination, the exposed area of the MCH(1 + 2) hydrogel undergoes self-regulated volume expansion, which in turn results in an increase in the curvature and a rightward shift in the center of gravity. Therefore, the O-ring hydrogel is able to reverse its rolling direction from left to right without any human control under the same constant illumination. Such bidirectional rolling motion with self-regulated direction reversal capability under constant light illumination was indeed observed in our experiments (Fig. 6c and Supplementary Movie 3). As a control, the single-component MCH2 hydrogel displayed monotonous unidirectional rolling toward the light source (Fig. 6b and Supplementary Movie 4), while the single-component MCH1 hydrogel was found to exhibit monotonous unidirectional rolling away from the light source (Fig. 6d and Supplementary Movie 5). These results further verified that by taking advantage of the specific structure/geometry (e.g., O-ring shape) and the self-regulated bidirectional volume change of our MCH(1 + 2) hydrogel, it is possible to develop a self-adaptive bidirectional locomotion (e.g., rolling) function subjected to constant light illumination without any complex external control.

## Discussion
In summary, we have reported the molecular design of structurally homogeneous hydrogel actuators with excellent self-regulated deformation reversal capability subjected to constant light illumination. Such hydrogels require a designed chemical composition of two coexisting spiropyran compounds that undergo a totally opposite charge variation under the same irradiation conditions, with one descending (MCH2 or MCH4) and the other ascending (MCH1 or MCH3). The coexistence of these two spiropyrans smartly generates a sequential positive-neutral-negative charge variation under constant light illumination, and this charge reversal at the molecular level allows self-adjusted bidirectional volumetric contraction-expansion of the structurally homogenous hydrogel at the macroscale. This self-regulated nonmonotonous response to constant illumination without any external assistance differs significantly from previously reported examples of photoresponsive hydrogels that usually display monotonous volume changes relying on either a sophisticated heterogeneous structural design or complex manipulation of nonuniform external stimuli. The driving force for self-regulated bidirectional actuation originates from photoisomerization-induced charge and

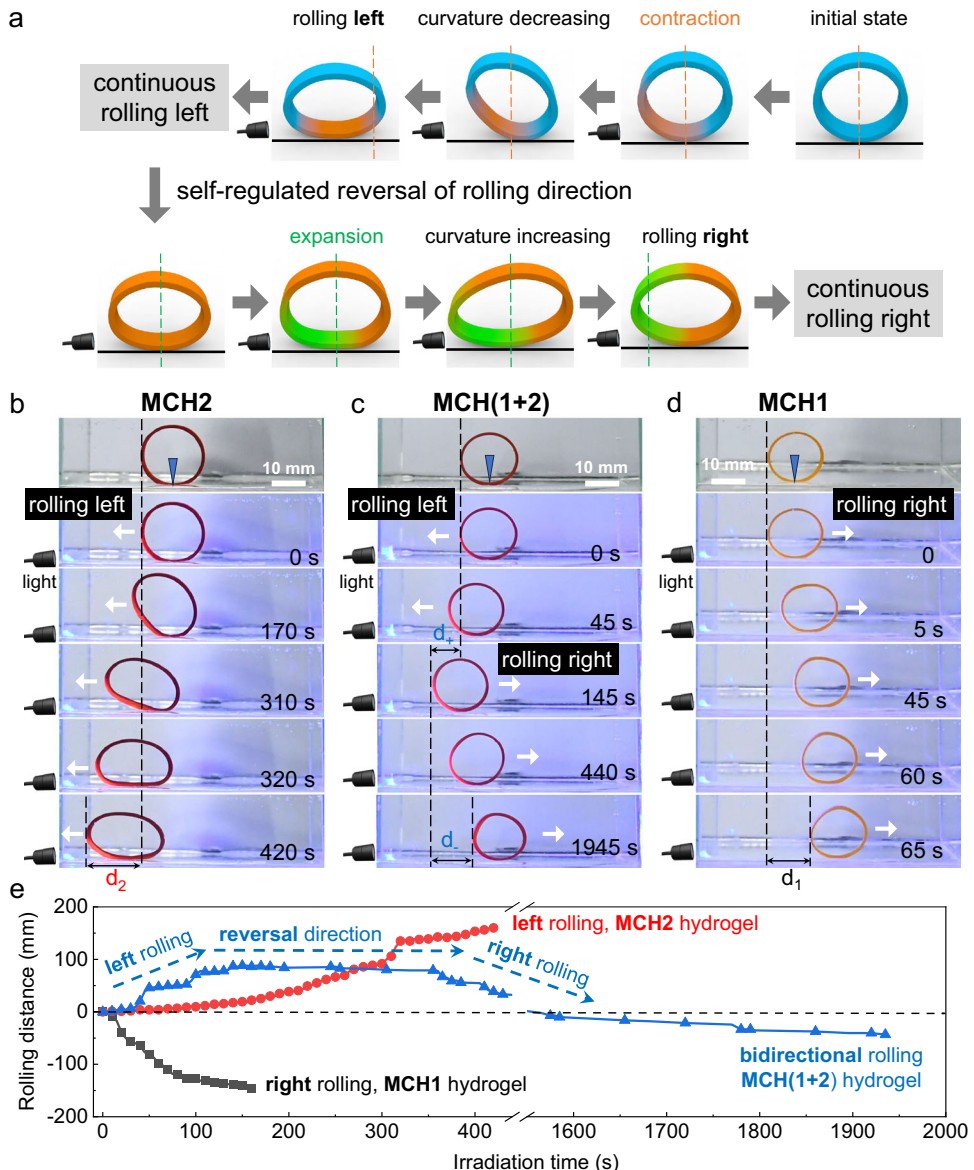

**Fig. 6 | Self-regulated reversal rolling motion. a** Schematic representation of the self-regulated bidirectional rolling motion of an O-ring-shaped MCH(1 + 2) hydrogel under constant irradiation from the left. **b** Photographs of an O-ring MCH2 hydrogel showing monotonous unidirectional rolling toward a light source giving a positive rolling distance of $d_2$. **c** Photographs of an O-ring-shaped MCH(1 + 2) hydrogel showing self-regulated nonmonotonous bidirectional rolling motion. The white arrow indicates the instant rolling direction; $d_+$ and $d_-$ represent the positive and negative rolling distance, respectively. **d** Photographs of an O-ring-shaped MCH1 hydrogel showing monotonous unidirectional rolling away from the light source giving a negative rolling distance of $d_1$. **e** Plot of the rolling distances of MCH1 (black), MCH2 (red), and MCH(1 + 2) hydrogels (blue) as a function of irradiation time under constant illumination (450 nm, 10.67 mW/cm²) from the left.

hydrophilicity variations, and there was no obvious photothermal effect involved (Supplementary Fig. 37). Based on the self-adjusted reversal volume change, hydrogel thin films were found to exhibit transient bidirectional bending deformation when subjected to constant illumination from one side. The bidirectional bending deformation phenomenon is widely observed in multiple hydrogel systems with variable combinations of four spiropyrans, including MCH(1 + 2), MCH(3 + 2) and MCH(3 + 4). Specifically, the MCH(1 + 2) (or MCH(3 + 4)) hydrogel has to contend with the light penetration depth due to their significant difference in photoisomerization speed, while the MCH(3 + 2) hydrogel can be considered a more uniform entity because MCH3 and MCH2 possess comparable slow photoisomerization speeds. These two models can generate bidirectional bending deformation when subjected to constant illumination, demonstrating the high robustness and strong universality of such a bidirectional deformation mechanism.

Furthermore, the actuation performance of bidirectionally deformable hydrogels could be flexibly tuned through precise control of the charge reversal behavior by selectively mixing two specific spiropyrans and varying the molar ratio or total grafting density of two combined spiropyrans, which was also found to be dependent on the environmental pH and irradiation intensity. Various bioinspired bidirectional shape transformations have been demonstrated to mimic or replicate certain features of motion/shape changes observed in nature. In addition, by taking advantage of the instant rolling of the O-ring shape as well as the self-adaptable bidirectional volume change of the MCH(1 + 2) hydrogel, our system exhibits promising self-regulated bidirectional rolling motion under constant illumination without any human control. Given the unique charge reversal design and robust bidirectional deformation mechanism subjected to constant illumination, our work represents an innovative strategy for programming nonmonotonous shape transformations and bidirectional locomotion

of homogeneous hydrogels using a single constant stimulus for future applications in intelligent actuators and soft robotics.

# Methods

## Materials

Chemicals including 4-methoxyphenylhydrazine hydrochloride (98%), *N,N,N′,N′*-tetramethylethylenediamine (99%), ammonium persulfate (98.5%), and *N,N′*-methylenebis(acrylamide) (AR) were purchased from Macklin. 3-methyl-2-butanone (99%), 2-bromoethanol (98%), triethylamine (99.5%), and methacryloyl chloride (97%) were purchased from Energy Chemical. 4-hydrazinylbenzenesulfonic acid hemihydrate (98%), 2,3,3-trimethylindolenine, salicylaldehyde (98%), 2-hydroxy-5-methoxybenzaldehyde (98%), 1,4-dioxane (99%), and *N*-isopropylacrylamide (98%), were purchased from Aladdin. Piperidine (AR) was purchased from Sinopharm Chemical Reagent. All other solvents (AR) were purchased from Chengdu Chron Chemical Co. Ltd.

## Synthesis and characterization of spiropyans

Compounds MCH1-MCH4 were synthesized according to Supplementary Fig. 1 following a procedure described in our previous work[40,42]. *Synthesis of 2,3,3-trimethyl-3H-indolium-5-sulfonate (1C).* A mixture of 4-hydrazinylbenzenesulfonic acid hemihydrate (15.0 g, 76.1 mmol) and 3-methyl-2-butanone (25.2 ml, 235.9 mmol) in glacial acetic acid (50 mL) was heated to 110 °C and refluxed for 3 h. Afterwards the mixture was slowly cooled to room temperature and acetic acid was removed with rotary evaporation. The resultant dark red oil was dissolved in MeOH (75 ml), followed by addition of saturated solution of potassium hydroxide in 2-propanol (100 ml) to get yellow solid precipitation, which was filtered and totally dried under vacuum (10.7 g, 59%). *Synthesis of compound B.* 1 equiv. of 1C or 2,3,3-Trimethylindolenine and 2 equiv. of 2-bromoethanol are dissolved in acetonitrile and stirred at 85 °C under reflux for 24 h. Afterward, the mixture is cooled to room temperature, and acetonitrile is removed by a rotary evaporator. The crude product is washed with Et₂O, filtered, and totally dried under vacuum to get pink solid 1B. After re-dissolved in dichloromethane and extracted with water, the aqueous phase is collected and rotary evaporated at 70 °C, and dried completely under a high vacuum to obtain red salt 2B. *General synthesis of spiropyran (A).* 1 equiv. of B and 1.2 equiv. of different methoxy-substituted salicylic aldehydes are added to ethanol, followed by addition of 1.2 equiv. of piperidine. The mixture was stirred at 60 °C for 0.5 h, followed by refluxing at 85 °C for 4 h. After cooling to room temperature and the solvent is removed by rotary evaporation. The product was purified by silica column using PE/DCM or DCM/MeOH as eluent to obtain the corresponding spiropyran. *Synthesis of MCH1-MCH4.* 1 equiv. of spiropyran (A) is dissolved in tetrahydrofuran, followed by addition of 2.5 equiv. of triethylamine, and the mixture is cooled to 0 °C. After dropwise addition of 2.5 equiv. of methacryloyl chloride, the solution was stirred at 0 °C for 1 h, and then at room temperature for 3 h. The solvent is removed by rotary evaporation and purified by a silica column using PE/DCM or DCM/MeOH as an eluent to obtain methacrylate-spiropyran. ¹H and ¹³C nuclear magnetic resonance (NMR) spectra were taken on a Bruker ASCEnd (TM) 400 MHz spectrometer at ambient temperature. The molecular weights were determined using high-resolution mass spectrometry (Acquity UPLC-Xevo G2 QTof).

## UV–Vis absorbance spectra

MCH1-MCH4 were separately dissolved in a solvent of methanol/water (4:1, v/v) containing 5 mM HCl to obtain a final concentration of 10 mM as a stock solution, followed by equilibration overnight in the dark at 25 °C. 20 μL of the stock solution was pipetted rapidly into 2000 μL of methanol/water (4:1, v/v) in a path length of a 10 mm removable quartz cuvette, followed by immediate irradiation with blue light (450 nm, 154.6 mW/cm²) and collection of absorbance spectroscopy using an Ocean Insight DH-2000-BD fiber spectrometer. The

photoisomerization rate under irradiation was obtained by plotting the characteristic absorbance of the MCH form (422.5 nm, 450.3 nm, 456.5 nm, and 422.5 nm for MCH1, MCH2, MCH3, and MCH4, respectively) vs irradiation time and fitting to the ExpDec1 function in Origin 2023 software. For the system containing mixed spiropyrans, the stock solutions were mixed first with variable ratios, followed by UV–vis measurements according to the same protocol described above. To determine the equilibrated absorbance spectra of the MCH isomers at different pH values, MCH1-MCH4 were dissolved in a solvent of methanol/water (4:1, v/v) containing different concentrations of HCl. The solution was incubated in the dark overnight before measurement.

## Calculation of the charge variation of mixed spiropyrans

By using the ExpDec1 function in Origin 2023 software to fit the plot of the absorbance of MCH form versus irradiation time, we obtained a mathematical expression of the ring-closing rate of MCH1 and MCH2, while the ring-closing rate represents the charge variation of each molecule; thus, the charge variation of MCH1 can be expressed as $Q_1 = -x[1 - (1.17342e^{k_1 t} - 0.03687)]$, the charge variation of MCH2 can be expressed as $Q_2 = (2-x)(0.9168e^{k_2 t} + 0.08073)$, the charge variation of MCH3 can be expressed as $Q_3 = -x[1 - (1.03062e^{k_3 t} - 0.01366)]$, the charge variation of MCH4 can be expressed as $Q_4 = (2-x)(1.20070e^{k_4 t} - 0.02912)$, where x reprensents the proportion of MCH1 (or MCH3) in the mixture and t represents the illumination time. The net charge variation of mixed MCH1 and MCH2 can be expressed as

$$Q_{netcharge} = Q_1 + Q_2 = -x[1 - (1.17342e^{k_1 t} - 0.03687)] + (2-x)(0.9168e^{k_2 t} + 0.08073)$$

The net charge variation of mixed MCH3 and MCH2 can be expressed as

$$Q_{netcharge} = Q_3 + Q_2 = -x[1 - (1.03062e^{k_3 t} - 0.01366)] + (2-x)(0.9168e^{k_2 t} + 0.08073)$$

The net charge variation of mixed MCH1 and MCH4 can be expressed as

$$Q_{netcharge} = Q_1 + Q_4 = -x[1 - (1.17342e^{k_1 t} - 0.03687)] + (2-x)(1.20070e^{k_4 t} - 0.02912)$$

The net charge variation of mixed MCH3 and MCH4 can be expressed as

$$Q_{netcharge} = Q_3 + Q_4 = -x[1 - (1.03062e^{k_3 t} - 0.01366)] + (2-x)(1.20070e^{k_4 t} - 0.02912)$$

After obtaining the mathematical expression of the net charge variation in mixed MCH1 and MCH2 (MCH3 and MCH2), MATLAB software was used to simulate the net charge variation versus irradiation time.

## Hydrogel preparation

Following the preparation formula listed in Supplementary Tables 1–6, a series of hydrogels containing mixed spiropyrans with different mixing ratios were prepared. In general, 10 wt% N-isopropylacrylamide (NIPAAm), 5 mol% (relative to the mole of the NIPAAm monomer), N,N-methylenebisacrylamide (MBAAm), and mixed spiropyrans were dissolved in a dioxane/water (4:1, v/v) mixture solvent (1 ml). The precursor solution was deoxygenated, followed by the addition of 50 μL of ammonium persulfate initiator (APS, 10 wt%) and 3.7 μL of tetramethylethylenediamine (TEMED). Free-radical polymerization was carried out at 25 °C for 4.5 h in a glass mold with a gap of 0.5 mm. After polymerization, the hydrogels were soaked in a large amount of DI

water to replace the organic solvent, followed by incubation in DI water containing 5 mM HCl overnight before use.

## Measurement of the change in hydrogel size under irradiation
A hydrogel dish (diameter of 8 mm and thickness of 0.5 mm) was irradiated simultaneously from both sides instead of via single-sided irradiation to avoid any bending deformation, as bending might affect the accuracy of the diameter change. The hydrogel size was defined as the change in diameter of the dish-shaped hydrogel during the light irradiation process, during which images were taken and measured using ImageJ software.

## Rheological measurements
To measure the mechanical changes in the hydrogel as a function of irradiation time, the hydrogel dishes were irradiated from both sides for different durations, after which the rheological properties were measured using a TA Discovery Hybrid Rheometer. Rheology experiments were performed in 8 mm parallel plates by using hydrogel films with a diameter of 8 mm and a thickness of 0.5 mm. A time scan was performed at a fixed frequency of 0.5 Hz and strain of 1.0% at 25 °C.

## Scanning electron microscopy (SEM) characterization
Hydrogels were prepared following the same protocol described above and incubated in DI water containing 5 mM HCl before measurement. Hydrogels before or after irradiation (450 nm, 160.1 mW/cm$^2$) for different durations were immediately frozen in liquid nitrogen and lyophilized, followed by coating with gold before observation using a scanning electron microscope (GeminiSEM500).

## Characterization of the bidirectional deformation of homogenous hydrogels
As an example, MCH(1 + 2) hydrogel ribbons (0.5 mm thick) were incubated in DI water containing 5 mM HCl in the dark overnight. The length and width of the hydrogel ribbons were fixed at 3 cm and 1 mm, respectively. The hydrogel ribbon was held vertically from the top with aluminum foil and immersed in a DI water tank, followed by immediate irradiation from the left with blue light (450 nm) using an optical goose neck (diameter of 6 mm) focusing on the central part of the ribbon. A digital camera was used to take a video from the front view to track the bidirectional bending process. We defined this as positive bending when the hydrogel ribbons moved toward the light, and in contrast, we defined this as negative bending when the hydrogel ribbons moved away from the light. ImageJ software was used to determine the bending angle, and the evolution of the bending angle over time was recorded. Each data point on the curve represents the average outcome of three individual experiments. Other hydrogel ribbons with variable compositions were characterized using the same protocol described above. To study the effect of light intensity on bidirectional bending, hydrogel ribbons in DI water were irradiated with variable light intensities (7.34, 12.00, 15.24, and 26.11 mW/cm$^2$). To investigate the pH effect of the water tank, the hydrogel ribbons were immersed in a water tank with variable pH values (2.09, 4.39, 6.60, and 7.24) followed by irradiation with a fixed light intensity of 15.24 mW/cm$^2$.

## Reversibility of bidirectional bending deformation
A star-shaped MCH(1 + 2) hydrogel film containing a fixed mixing ratio of 1.0:1.0 (MCH1:MCH2) and a fixed total grafting density of 2.0% was used for testing reusability. After incubation in DI water containing 5 mM HCl in the dark overnight, the whole sample was immersed vertically into a water tank containing DI water, followed by irradiation with blue light from left to start the bidirectional process. Once a cycle was finished, the sample was incubated in DI water containing 5 mM HCl in the dark before the next test.

## Bidirectional rolling motion
O-ring-shaped hydrogels were prepared by adding the monomer solution to a plastic chamber mold composed of two nested plastic O-rings, followed by polymerization to form the hydrogel and demolding. The thickness of the hydrogel wall and the height of the O-ring hydrogel were fixed at 1.0 mm and 5.0 mm, respectively. The O-ring hydrogels were placed in DI water containing 5 mM HCl in the dark overnight before being transferred to a DI water tank and irradiated from the left with blue light (450 nm, 10.67 mW/cm$^2$). A digital camera was used to take a video from the front view to track the bidirectional rolling process.

## Data availability
The data that support the findings of this study are available within this article (and its Supplementary Information files) and from the corresponding authors upon request.

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

## Acknowledgements

This work acknowledges the financial support from the National Natural Science Foundation of China (52373121), the National Key R&D Program of China (2022YFA1305100), the Natural Science Foundation of Anhui Province (2208085MB27) and the University of Science and Technology of China (KY2060000212). We thank Mengqi Du for help with the cartoon illustration in Fig. 6a.

## Author contributions

K.G. performed the experiments. X.Y. participated in part of the experiments. C.Z. participated in the discussion. C.L. supervised the research and wrote the manuscript.

## Competing interests

The authors declare no competing interests.
