## [Peer Review File · Nature Communications]

REVIEWER COMMENTS

Reviewer #1 (Remarks to the Author):

The authors have developed a homogenous hydrogel system integrated with two coexisting photoswitchable spiropyrans, which can achieve bidirectional bending under constant light. The phenomenon is unique, and the relevant demonstration presents great controllability in practice. However, one of the key mechanisms has not been explained clearly and the experiment result seems to be the opposite of the theory and calculation. That is, the MCH1 part will be triggered first and expand, while the hydrogel will contract instead. Overall, this work is still worthwhile, and it can be considered to be published with major revisions and deeper understanding. Some specific comments are listed below.

1. Please further explain why the neutral state leads to the expansion while the negatively or positively charged state causes the contraction. It is not clear on page 7, which asks for the support of theory and references at least.
2. How about the structural change between the ring-opening MCH and ring-closed SP, which will also generate the spatial difference in addition to the charge influence?
3. What is the exact synergistic interaction to cause the opposite result vs the simulation result in Figure 1d? It seems the MCH1 will be triggered first, leading to the expansion of the hydrogel, followed by the contraction based on MCH2 part. The current description is not clear and should be further investigated.
4. What is the definition of the hydrogel size and how did you test the hydrogel size?
5. As for Figure 2c, how about the basic solution with the $\text{pH} > 7$?
6. From Figure S4 and Figure 1d, the illuminated MCH1 will expand with the pore size increasing. Still, why will it lead to the contraction in Figure 2 and "negative bending angles" on page 10 related to Figure 2e?
7. Please double-check the details, e.g. missing the "min" in Figure 5a.

Reviewer #2 (Remarks to the Author):

The paper, by Li, et al. reports a very interesting photochemical hydrogel system:

First, the system is chemically homogeneously distributed and is designed for the excitation upon a non-modulated, constant, non-coherent light field. There is no human control of the light field, referred to as "material self-regulation".

Second, there is actuation (contraction/expansion, bending) reversal, indicating the presence of a built-in intermediate state in the responsive system. This actuation behavior is unusual.

Third, the mechanism is based on the competition between two different spiropyran photoswitches – one photoisomerizes faster than the other, contributing to different charges. This opens up many new actuation possibilities, simply by changing the isomer mixture. The authors have also demonstrated changes in behavior by varying grafting percentage, light intensity, and pH in different types of hydrogel systems (1+2, 2+3, Fig. 2, Fig. 4).

Fourth, the actuator demonstration, especially the reversal of rolling direction in Fig. 6, is very impressive.

This work reminds me one paper, “Zhao, et al. Desynchronized liquid crystalline network actuators with deformation reversal capability”, in which the liquid crystal elastomer bending reversal was reported by using purely physical approach. This work reports the deformation reversal through purely chemical approach.

It is quite evident that the hydrogel gel actuator responds very slowly (all videos are accelerated). While it is understandable that critics might point out the sluggish response in real-world applications. From a biomimetic standpoint, many natural species undergo very slow shape-morphing.

So, the speed is not a significant concern in this basic research work, but, this is my personal perspective.

Minor comments:

1. When discussing about the intelligent adaptation of natural species, the authors listed Ref.1, and mentioned, “reversible movement of pinecone scales in response to changes in humidity is attributed to the dynamic reorientation of internal cellulose microfibrils within cellular walls.” If I am not wrong, I did not see any reversible humidity driven deformation in pinecone from that Ref.
2. Fig. 1d, 3d, the calculation curve. I suggest that the authors replace it with a dashed line to distinguish it from the actual experimental data.
3. In Fig. 2g and 4g, there is no indication of illumination time. I believe the dose/exposure time is crucial to reaching the photostationary state.
4. It took me quite some time to appreciate the two different models, MCH(1+2) and MCH(2+3). Due to the difference in isomerization speed, one of these models has to contend with penetration depth, while the other can be considered to be a more uniform entity. The deformation behaviors are also distinct, and this is convincing.
5. I would like to request the authors to provide two comments or engage in a brief discussion:
 - (i) In Fig. 6b, the reversed direction in rolling motion is noteworthy. However, could the authors elaborate on the conditions that might lead to failure in this experiment (considering high buoyancy, unexpected soft adhesive, presence of bubbles, etc)?
 - (ii) Regarding the introduction of more oscillation cycles in the deformation (as seen in, for example, Fig. 2b and S9e, all show one cycle of oscillation), could the authors explore achieving this solely through

photoswitches induced net-charge competition, without involving physical self-shadowing (doi.org/10.1002/adfm.202214394)? My initial intuition suggests it may not be possible. Perhaps the authors could share their perspective or offer other insights.

Reviewer #3 (Remarks to the Author):

Hydrogels with shape deformability and locomotive capability in response to environmental stimulation are of great importance for the development of bioinspired intelligent actuators, functional sensors and soft robots. A grand challenge in this field is to design and synthesize self-adaptive hydrogels that are able to autonomously sense, adjust and adapt to external changes without any human interference or control. This manuscript entitled "Self-regulated reversal deformation and locomotion of structurally homogenous hydrogels subjected to constant light illumination" reports a useful strategy to endow structurally homogenous hydrogels with self-regulated reversal deformation and locomotion capability when subjected to constant illumination by a rational design and combination of different spiropyran photoswitches. Based on these advantages, together with that the experiments are well conducted, the paper is well-organized, and the topic matches well with the scope of the journal, I recommend for a publication after addressing the following minor issues:

(1) This is a very unique photoactuator system, where light-induced charge reversal of spiropyrans at the molecular level was smartly utilized to sequentially generate an opposite volume change (contraction-expansion) at the macroscale, resulting in a series of interesting self-regulated bidirectional shape deformation behaviors under constant illumination. Currently, there are numerous papers in the literature that study photoactuators, the authors should clearly demonstrate the advantages of this design compared with other systems.

(2) The complex bidirectional deformation was generated by the hydrogels themselves through a feedback of charge reversal in response to constant external illumination, which does not require any other human control. How to precisely control such charge reversal, specially when the stimulus is not constant?

(3) The photoresponsive hydrogels investigated in the paper are structurally homogenous, without any fixed layered/oriented heterogeneous structures, which are dynamic and allow for a highly reversible deformation. In general, compared with heterogeneous actuators, is there any disadvantages of the homogenous hydrogels?

(4) The light intensity used in different systems ranged from tens to hundreds of mW/cm². The authors should check whether there is any photothermal effect during the photoactuation experiments.

(5) What is the real scale of the dual-star-shaped hydrogel shown in Figure 5c? A scale bar is missing and needs to add. The same issue occurred for the hydrogel trips shown in Figure 2b and 4b.

Response to reviewers' comments

Reviewer #1

The authors have developed a homogenous hydrogel system integrated with two coexisting photoswitchable spiropyrans, which can achieve bidirectional bending under constant light. The phenomenon is unique, and the relevant demonstration presents great controllability in practice. However, one of the key mechanisms has not been explained clearly and the experiment result seems to be the opposite of the theory and calculation. That is, the MCH1 part will be triggered first and expand, while the hydrogel will contract instead. Overall, this work is still worthwhile, and it can be considered to be published with major revisions and deeper understanding.

Our response: We appreciate this reviewer for the thorough review and positive comments on our work.

We apologize for confusing the reviewer on the understanding of the actuation mechanism. Here, we concisely respond to this question, as we have provided a more detailed explanation/clarification in the following point-by-point response. The confusion arises from a conceptual misunderstanding; that is, the change in volume of the mixed MCH(1+2) hydrogel under irradiation is a simple physical sum of that of the individual MCH1 and MCH2 hydrogels, which is apparently not the case in our work. In fact, the change in volume of the mixed MCH(1+2) hydrogel upon irradiation is determined simultaneously by the total net charge change in both the MCH1 and MCH2 compounds, in which case these two spiropyrans should not be considered separately/individually but rather synergistically as a whole. Therefore, although MCH1 is triggered first, this change does not lead to volume expansion as does the single-component MCH1 hydrogel; instead, it results in volume contraction of the mixed MCH(1+2) hydrogel. This is because the coexistence of the neighboring positively charged MCH2 quenched/neutralized the negative charge of SP1, leading to a total net charge decrease and volume contraction of the mixed MCH(1+2) hydrogel. This result verified that the change in volume of the mixed MCH(1+2) hydrogel under the same constant irradiation conditions significantly differed (or behaved oppositely) from that of the single-component MCH1 hydrogel. We have added more detailed clarifications and text modifications in response to comment 1, 3 and comment 6 regarding this confusion.

Some specific comments are listed below.

1. Please further explain why the neutral state leads to the expansion while the negatively or positively charged state causes the contraction. It is not clear on page 7, which asks for the support of theory and references at least.

Our response: We apologize for the confusion. The volume change (either expansion or contraction) of the spiropyran hydrogel in response to light depends directly on the net charge variation of the spiropyran molecules upon photoisomerization. The positive correlation between the change in hydrogel volume and the variation in the spiropyran charge was verified by both experiments and simulations in our previous work (JACS, 2020, ref 42) and can be summarized briefly as follows: an increase (or decrease) in the net charge after photoisomerization enhances the hydrophilicity (or hydrophobicity) of the polymer chains, therefore leading to a volume expansion (or contraction) of the hydrogel. This principle applies to the single-component MCH1 and MCH2 hydrogels, where the MCH1 hydrogel expands because the charge increases upon irradiation, while the MCH2 hydrogel contracts as the charge decreases after irradiation (see the following Response Figure 1). This principle also applies to the mixed MCH(1+2) hydrogel described in this work, whose volume change in response to light depends simultaneously on the total net charge change in the MCH1 and MCH2 compounds. In this case, the coexisting MCH1 and MCH2 are treated synergistically as a whole, and the total charge state of the MCH(1+2) hydrogel undergoes a sequential variation of “positive-neutral-negative” under constant light illumination. The transition from the initial positively charged state to the neutral state at the beginning leads to a volume contraction of the MCH(1+2) hydrogel due to the decrease in the total net charge, while the latter transition from the neutral to negatively charged state with prolonged irradiation leads to a volume expansion

of the MCH(1+2) hydrogel due to the increase in the total net charge. Therefore, our mixed MCH(1+2) hydrogel shows sequential contraction-expansion in response to constant light irradiation.

Response Figure 1 (for review only). Illustration of the net charge and volume change of single-component **MCH1** (a) and **MCH2** (b) as well as the mixed **MCH(1+2)** hydrogel (c) before and after irradiation.

To clarify this point more clearly, we added a discussion on page 7 of the revised manuscript, stating “The transition from the initial positively charged state to the transient neutral state is expected to lead to a volumetric contraction of the **MCH(1+2)** hydrogel because a decrease in the total net charge lowers the hydrophilicity of the polymer chains, which would drive water molecules to diffuse out of the hydrogel.⁴²”, and “Volumetric expansion is expected to occur in the transition process from the transient neutral state to the final negatively charged state as an increase in the total net charge enhances the hydrophilicity of the polymer chains and drives the diffusion of water molecule into the hydrogel.⁴²”. **Page 8, we state** “Therefore, hydrogel contraction or expansion upon irradiation directly depends on the total net charge decrease or increase in imbedded spiropyran,⁴² and this principle not only applies to single-component **MCH1** and **MCH2** hydrogels but also applies to two-component **MCH(1+2)** hydrogels in which the total net charge of two spiropyrans plays a decisive role.”

2. How about the structural change between the ring-opening MCH and ring-closed SP, which will also generate the spatial difference in addition to the charge influence?

Our response: In the literature, it is known that light-induced charge variation and hydrophilicity changes determine the volume change of spiropyran hydrogels (*Chem. Mater.* 2007, 19, 2730; *Soft Matter* 2011, 7, 8030; *J. Am. Chem. Soc.* 2020, 142, 8447). The spatial difference between the ring-opening MCH and ring-closed SP plays a minor role in changing the volume of the hydrogel. This is because the spiropyran molecule is appended

(not crosslinked) to the polymer chains, and its structural change tends to dissipate randomly. Therefore, it is very difficult to transfer spatial differences to polymer chains to induce any macroscale volume change.

To directly test the individual influence of spatial differences on the change in volume of the spiroopyran hydrogel, we synthesized a new nitro-spiroopyran that can be easily ring-opened by UV light in DI water to obtain a nonprotonated merocyanine isomer (MC⁻; see the following Response Figure 2a-2c). In this way, the charge states of the SP and MC⁻ isomers could be treated the same (the SP is neutral, and the MC⁻ is zwitterionic), while their spatial geometries differ, providing us with an opportunity to investigate only the spatial influence. We found that the prepared nitro-spiroopyran hydrogel displayed an obvious color change upon irradiation with UV light (365 nm; see Response Figure 2d), indicating the successful generation of the ring-opening MC⁻ form. However, no obvious change in volume was observed during this process. This result directly verified that the spatial change in the spiroopyran molecule cannot result in an obvious change in the volume of the hydrogels, which could be attributed to the fact that the appended spiroopyrans reconfigure freely and are unable to effectively transfer their geometric changes to the macroscale volume change of the hydrogel.

Response Figure 2 (for review only). (a) Chemical structure of nitro-spiroopyran. The color change (b) and UV-vis absorption spectrum (c) of nitro-spiroopyran in a solvent of methanol:water = 4:1 (v/v) upon irradiation at 365 nm and 450 nm. (d) Photographs of a dished nitro-spiroopyran hydrogel upon irradiation at 450 nm and 365 nm.

3. What is the exact synergistic interaction to cause the opposite result vs the simulation result in Figure 1d? It seems the MCH1 will be triggered first, leading to the expansion of the hydrogel, followed by the contraction based on MCH2 part. The current description is not clear and should be further investigated.

Our response: As clarified in response to comment 1, the change in volume of the mixed MCH(1+2) hydrogel upon irradiation was determined simultaneously by the total net charge change in both the MCH1 and MCH2 compounds; in these cases, these two spiroopyrans were not considered separately/individually but rather should be treated synergistically as a whole. The total charge state of the MCH(1+2) hydrogel undergoes a sequential variation of “positive-neutral-negative” under constant light illumination, leading to a sequential volume

contraction-expansion of our mixed MCH(1+2) hydrogel. In this bidirectional volume change process, although MCH1 is triggered first, it will not induce photoexpansion behavior similar to that observed in the single-component MCH1 hydrogel; instead, we observed opposite photocontraction behavior in the mixed MCH(1+2) hydrogel. This is because the coexistence of neighboring positively charged MCH2 (+1) neutralizes or eliminates the negative charge of SP1. For example, before irradiation, the total net charge of MCH(1+2) is positively charged [MCH1 (0) + MCH2 (+1) = +1]; upon short-term irradiation to trigger MCH1 isomerization, MCH2 remains unchanged, and the total net charge of MCH(1+2) becomes neutral [SP1 (-1) + MCH2 (+1) = 0], leading to volume contraction because the charge decreases from +1 to 0; similarly, the latter isomerization of MCH2 to SP2 does not lead to a volume contraction like that of the single-component MCH2 hydrogel but instead leads to a volume expansion of the mixed MCH(1+2) hydrogel, as the total net charge state changes from a neutral to a negatively charged state [SP1 (-1) + MCH2 (0) = -1]. In other words, the first contraction of the mixed MCH(1+2) hydrogel is actually triggered by the isomerization of MCH1 (MCH1 isomerized to SP1; MCH2 kept unchanged), while the subsequent expansion is actually induced by the isomerization of MCH2 (SP1 kept unchanged; MCH2 isomerized to SP2). These results indicated that the roles of MCH1 and MCH2 in the mixed MCH(1+2) hydrogel are opposite to those in the single-component MCH1 and MCH2 hydrogels. The simulation results in Figure 1d were obtained by averaging the expansion value of the MCH1 hydrogel and the contraction value of the MCH2 hydrogel; these results are opposite to those obtained for our experimental MCH(1+2) hydrogel.

To clarify this point more clearly, we added a discussion on pages 7-8 of the revised manuscript stating “Specifically, although the single-component MCH1 hydrogel itself displayed a volume expansion upon irradiation, the isomerization of MCH1 to SP1 in the mixed MCH(1+2) hydrogel did not lead to an expansion but instead gave an opposite volume contraction due to the simultaneous presence of positively charged MCH2 quenching/neutralizing the negative charge of SP1. Similarly, the isomerization of MCH2 to SP2 does not generate a volume contraction like a single-component MCH2 hydrogel but instead leads to a volume expansion of the mixed MCH(1+2) hydrogel due to the presence of SP1. The role that MCH1 (or MCH2) plays in the mixed MCH(1+2) hydrogel is opposite to that in the individual MCH1 (or MCH2) hydrogel, generating a calculation curve of volume change with the opposite trend, as shown in Figure 1d. These results indicated that the nonmonotonous contraction-expansion phenomenon observed in our MCH(1+2) hydrogel is rooted in the synergistic interaction between MCH1 and MCH2, which should be considered not separately but synergistically as a whole. Therefore, hydrogel contraction or expansion upon irradiation directly depends on the total net charge decrease or increase in imbedded spiropyran,⁴² and this principle not only applies to single-component MCH1 and MCH2 hydrogels but also applies to two-component MCH(1+2) hydrogels, in which the total net charge of two spiropyrans plays a decisive role.”

4. *What is the definition of the hydrogel size and how did you test the hydrogel size?*

Our Response: We apologize for the missing information on the definition of hydrogel size in our first version of the manuscript. The hydrogel size was defined as the change in the diameter of the dish-shaped hydrogel with increasing light irradiation. In this case, the hydrogel dish was irradiated simultaneously from both the top and the bottom sides (instead of via single-sided irradiation) to avoid any bending deformation, as bending might affect the accuracy of the diameter change. Photographs were taken during the irradiation process, and the change in diameter was measured using ImageJ software. This detailed information has been added to the captions of Figures 1d and 3d, stating “Plot of hydrogel size (% , defined as the percentage change in diameter of the dish-shaped hydrogel) as a function of two-sided irradiation (8.99 mW/cm²) time”. We also added an experimental description on page 6 of the revised supporting information, stating “Measurement of the change in hydrogel size under irradiation. A hydrogel dish (diameter of 8 mm and thickness of 0.5 mm) was irradiated simultaneously from both sides instead of via single-sided irradiation to avoid any bending deformation, as bending might affect the accuracy of the diameter change. The hydrogel size was defined as the change in diameter of the dish-shaped hydrogel during the light irradiation process, during which images were taken and measured using ImageJ software.”

5. As for Figure 2c, how about the basic solution with the $pH > 7$?

Our response: We appreciate the reviewer for bringing up this question. New experiments were carried out to investigate the effect of basic solution on the photoactuation behaviors of our hydrogels. Using the same hydrogel ribbon (30 mm long, 1 mm wide and 0.5 mm thick) under the same irradiation setup (blue light, 15.24 mW/cm^2) but using a basic water tank ($pH = 7.24$), we observed similar bidirectional bending behaviors for both the MCH(1+2) hydrogel and the MCH(3+2) hydrogel under constant irradiation. These results are in good agreement with our clarification on page 10 of the previous version of our manuscript, stating “a relatively high pH facilitates the creation of such a reversal deformation due to the spontaneous deprotonation and ring-closure of MCH at higher pH (Figure S5)”. These results further verified that the light-driven bidirectional bending phenomenon of our hydrogels could be widely observed in various pH environments, demonstrating the robustness of the bidirectional bending mechanism. These new results have been added to the revised Figure 2c and Figure 4c in the revised manuscript.

Revised Figure 2c (left, MCH(1+2) hydrogel) and **4c** (right, MCH(3+2) hydrogel) with an added bending curve (green) under a basic environment.

6. From Figure S4 and Figure 1d, the illuminated MCH1 will expand with the pore size increasing. Still, why will it lead to the contraction in Figure 2 and “negative bending angles” on page 10 related to Figure 2e?

Our response: The reason why MCH1 isomerization triggers contraction of the MCH(1+2) hydrogel in Figure 2 has been clarified clearly in detail in our previous response to comments 1 and 3 (please see above).

In terms of the bending performance, the bending direction (positive or negative bending) and the bending angle are directly determined by the light-induced contraction or expansion gradient throughout the thickness of the hydrogel ribbon and are dependent on the molar ratio of MCH1:MCH2. When the molar ratio is fixed at 1:1, the isomerization of MCH1 to SP1 leads to a contraction gradient for a positive bend toward the light source; later, the photoisomerization of MCH2 to SP2 gives an expansion gradient and a negative bend. The positive and negative bending angles in this case with a 1:1 ratio is close to symmetric. However, when the ratio deviates from 1:1, the positive and negative bending angles are found to become asymmetric as well. We found that a higher molar ratio of MCH1 leads to a greater negative bending angle (Figure 2e). A higher molar ratio of MCH1 induces a greater absolute value of charge change from the neutral to negative state (Figure 2d), corresponding to the photoexpansion process for negative bending. For example, for a molar ratio of MCH1:MCH2 = 1.5:0.5, the net charge of the MCH(1+2) hydrogel would undergo a sequential change of $+0.5 \rightarrow 0 \rightarrow -1.5$ upon irradiation, in which the former change of $+0.5 \rightarrow 0$ determines the contraction gradient for positive bending, while the latter change of $0 \rightarrow -1.5$ determines the expansion gradient for negative bending. Because the absolute value of $0 \rightarrow -1.5$

in the latter process is 1.5, which is larger than the 0.5 of $+0.5 \rightarrow 0$, the negative bending angle is much larger than the positive bending angle. In contrast to the case for a molar ratio of $\text{MCH1}:\text{MCH2} = 0.5:1.5$, the net charge of the $\text{MCH}(1+2)$ hydrogel underwent a sequential change of $+1.5 \rightarrow 0 \rightarrow -0.5$ upon irradiation, which resulted in a larger positive bending angle but a smaller negative bending angle. These results indicated that although the photoisomerization of MCH1 leads to a contraction gradient for positive bending deformation (this is from the perspective of the bending direction), a higher molar ratio of MCH1 in the mixed $\text{MCH}(1+2)$ hydrogel instead generates a larger negative bending angle (this is from the perspective of the bending angle).

Revised Figure 2 Figure 2d (Left), plot of the net charge changes in mixed MCH1 and MCH2 with variable ratios as a function of irradiation time, showing positively and negatively charged regions with tunable charge transition time points. Right, Figure 2e, plot of the bending angles of $\text{MCH}(1+2)$ hydrogel ribbons with variable mixing ratios under constant irradiation (450 nm , 15.24 mW/cm^2).

To clarify this point more clearly, we added a discussion on page 11 of the revised manuscript stating “The former process from the initial positive charge state to the neutral state under irradiation corresponds to the contraction gradient for a positive bending deformation, while the latter process from the neutral to negative charge state governs the expansion gradient for a negative bending deformation. A molar ratio of 1:1 results in symmetric positive and negative bending angles under constant irradiation. However, when the ratio deviates from 1:1, a higher molar ratio of MCH1 results in a shorter positive-negative charge reversal time and a greater absolute value of negative charge change in the mixed $\text{MCH}(1+2)$ hydrogel (Figure 2d)”, and “It was found that a higher molar ratio of MCH1 (or MCH2) produces a larger negative (or positive) bending angle due to the total greater negative (or positive) charge change under irradiation.”

7. Please double-check the details, e.g. missing the “min” in Figure 5a.

Our response: We thank the reviewer for pointing out this issue, and we have corrected it in the revised Figure 5a.

Reviewer #2:

The paper, by Li, et al. reports a very interesting photochemical hydrogel system:

First, the system is chemically homogeneously distributed and is designed for the excitation upon a non-modulated, constant, non-coherent light field. There is no human control of the light field, referred to as “material self-regulation”. Second, there is actuation (contraction/expansion, bending) reversal, indicating the presence of a built-in intermediate state in the responsive system. This actuation behavior is unusual.

Third, the mechanism is based on the competition between two different spiropyran photoswitches – one photoisomerizes faster than the other, contributing to different charges. This opens up many new actuation possibilities, simply by changing the isomer mixture. The authors have also demonstrated changes in behavior by varying grafting percentage, light intensity, and pH in different types of hydrogel systems (1+2, 2+3, Fig. 2, Fig. 4).

Fourth, the actuator demonstration, especially the reversal of rolling direction in Fig. 6, is very impressive. This work reminds me one paper, “Zhao, et al. Desynchronized liquid crystalline network actuators with deformation reversal capability”, in which the liquid crystal elastomer bending reversal was reported by using purely physical approach. This work reports the deformation reversal through purely chemical approach.

Our response: We are grateful to the reviewer for the thorough review of our paper and for the positive comments.

It is quite evident that the hydrogel gel actuator responds very slowly (all videos are accelerated). While it is understandable that critics might point out the sluggish response in real-world applications. From a biomimetic standpoint, many natural species undergo very slow shape-morphing. So, the speed is not a significant concern in this basic research work, but, this is my personal perspective.

Our response: We thank this reviewer for the comments on the actuation speed. Apparently, the aim of this work was not to develop hydrogel actuators with fast actuation speeds. Instead, our goal is to provide a new molecular design strategy for the development of intelligent hydrogel actuators with self-regulation ability that can autonomously change their deformation/locomotion direction when subjected to constant illumination. As the reviewer mentioned, many natural species undergo very slow shape morphing, so a fast speed is not always needed in real-world applications. In the future, this could be realized either by downscaling the actuator size to the micrometer scale or by introducing a porous structure to facilitate water diffusion. However, this is beyond the scope of this manuscript.

Minor comments:

1. When discussing about the intelligent adaptation of natural species, the authors listed Ref.1, and mentioned, “reversible movement of pinecone scales in response to changes in humidity is attributed to the dynamic reorientation of internal cellulose microfibrils within cellular walls.” If I am not wrong, I did not see any reversible humidity driven deformation in pinecone from that Ref.

Our response: We apologize for citing an incorrect reference in our first version of the manuscript. The correct reference for this statement should be Ref. 4, which describes the reversible opening and closing of pinecone in response to humidity as follows: “The passive opening and closing of the scales of a pinecone is an elegant example of hygroscopic actuation for seed dispersal, accompanied by a simple synthetic model. The reversible movement of the pine cone scales is driven by differences in structural orientation of the cellulose microfibrils within cell walls across the structure. The microfibrils have a greater resistance to extension along their axis of alignment, and so by varying this angle relative to the body of the cell, deformation of the volume can be channelled preferentially in one direction. Cells making up the outside of the scale are orientated to elongate on exposure to humidity, whereas the inner layer are more resistant to elongation”. We have now replaced Ref. 1 with the correct Ref. 4 in the revised manuscript.

2. Fig. 1d, 3d, the calculation curve. I suggest that the authors replace it with a dashed line to distinguish it from the actual experimental data.

Our response: Thank you for the good suggestion. We have now changed the calculation curve to a dashed line to distinguish it from the experimental data in the revised manuscript.

Revised Figure 1d (left) and 3d (right) with a dashed line to represent the calculation curve.

3. In Fig. 2g and 4g, there is no indication of illumination time. I believe the dose/exposure time is crucial to reaching the photostationary state.

Our response: Agreed. We added the detailed illumination time point at which the hydrogel ribbon reached its maximum positive and negative bending angles to Figure 2g and Figure 4g, respectively, in the revised manuscript. We also added a description to the corresponding figure caption, stating “The labeling time represents the illumination time when the hydrogel ribbon reaches its maximum bending angle.”

Revised Figure 2g (left) and 4g (right) with labeled illumination time when the maximum bending angles were reached.

4. It took me quite some time to appreciate the two different models, MCH(1+2) and MCH(2+3). Due to the difference in isomerization speed, one of these models has to contend with penetration depth, while the other can be considered to be a more uniform entity. The deformation behaviors are also distinct, and this is convincing.

Our response: Thank you for the positive comments. To better clarify and help understand the differences between these two models, we have added a short discussion to the conclusion section on pages 23-24 of the revised manuscript, stating “Specifically, the MCH(1+2) (or MCH (3+4)) hydrogel has to contend with light penetration depth due to their significant difference in photoisomerization speed, while the MCH(3+2) hydrogel can be considered a more uniform entity because MCH3 and MCH2 possess a comparable slow photoisomerization speed. These two models can generate bidirectional bending deformation when subjected to constant illumination, demonstrating the high robustness and strong universality of such a bidirectional deformation mechanism.”

5. I would like to request the authors to provide two comments or engage in a brief discussion:

(i) In Fig. 6b, the reversed direction in rolling motion is noteworthy. However, could the authors elaborate on the conditions that might lead to failure in this experiment (considering high buoyancy, unexpected soft adhesive, presence of bubbles, etc)?

Our response: As the driving force for the rolling motion of the O-ring-shaped hydrogel comes from the deformation-induced shift in the center of gravity, the key to obtaining the reversal rolling motion lies in how to transform the light-induced contraction-expansion deformation to the opposite shift in the center of gravity of the O-ring. Therefore, both the hydrogel parameters and the irradiation conditions play important roles in this experiment. First, the geometry and size of the hydrogel O-ring should match well with the light irradiation spot area (we used an optical gooseneck with a diameter of 5 mm) to create the desired deformation curvature for gravity center shifting upon irradiation. In our experiment, the O-ring diameter is 15 mm, the height is 5 mm, and the thickness of the wall is 1 mm. An O-ring that is either larger or smaller in size is more difficult to use for reversing rolling motion. Second, the irradiation conditions also matter. A suitable irradiation intensity (450 nm, 10.67 mW/cm²) and a right irradiation position (parallelly from the left and focused on the left-bottom area of the O-ring) are needed to guarantee the occurrence of the reversal rolling motion. Neither a stronger nor weaker light intensity nor an improper irradiation position will generate a suitable and continuous contraction and expansion gradient for opposite shifts in the gravity center, which will likely lead to failure. Any other possible distraction factors, such as adhesion from the substrate or the presence of bubbles, should also be avoided in this experiment.

(ii) Regarding the introduction of more oscillation cycles in the deformation (as seen in, for example, Fig. 2b and S9e, all show one cycle of oscillation), could the authors explore achieving this solely through photoswitches induced net-charge competition, without involving physical self-shadowing (doi.org/10.1002/adfm.202214394)? My initial intuition suggests it may not be possible. Perhaps the authors could share their perspective or offer other insights.

Our response: This is a very constructive suggestion! Currently, in this work, we are able to cycle the bidirectional deformation actuation of our hydrogels by manually switching the external light on and off (Figure S34). It would be highly important and interesting if oscillation behavior could be achieved solely through photoswitch-induced net-charge competition without involving a physical self-shadowing negative feedback loop. This is of course very challenging, and currently, we have no clear answer to this question. This might be a long-term goal that deserves continuous pursuit in future research. We appreciate the reviewer for this valuable suggestion.

Reviewer #3:

Hydrogels with shape deformability and locomotive capability in response to environmental stimulation are of great importance for the development of bioinspired intelligent actuators, functional sensors and soft robots. A grand challenge in this field is to design and synthesize self-adaptive hydrogels that are able to autonomously sense, adjust and adapt to external changes without any human interference or control. This manuscript entitled "Self-regulated reversal deformation and locomotion of structurally homogenous hydrogels subjected to constant light illumination" reports a useful strategy to endow structurally homogenous hydrogels with self-regulated reversal deformation and locomotion capability when subjected to constant illumination by a rational design and combination of different spiropyran photoswitches. Based on these advantages, together with that the experiments are well conducted, the paper is well-organized, and the topic matches well with the scope of the journal, I recommend for a publication after addressing the following minor issues:

Our response: We are very grateful to the reviewer for the thorough review of our paper and for the positive comments.

(1) This is a very unique photoactuator system, where light-induced charge reversal of spiropyran at the molecular level was smartly utilized to sequentially generate an opposite volume change (contraction-expansion) at the macroscale, resulting in a series of interesting self-regulated bidirectional shape deformation behaviors under constant illumination. Currently, there are numerous papers in the literature that study photoactuators, the authors should clearly demonstrate the advantages of this design compared with other systems.

Our response: We appreciate the reviewer's positive comments and useful suggestion. To better clarify the advantages of our design compared with those of other photoactuators in the literature, we have added a short discussion to the conclusion section of the revised manuscript, stating "The coexistence of these two spiropyran smartly generates a sequential "positive-neutral-negative" charge variation under constant light illumination, and this charge reversal at the molecular level allows self-adjusted bidirectional volumetric contraction-expansion of the structurally homogenous hydrogel at the macroscale. This self-regulated nonmonotonous response to constant illumination without any external assistance differs significantly from previously reported examples of photoresponsive hydrogels that usually display monotonous volume changes relying on either a sophisticated heterogeneous structural design or complex manipulation of nonuniform external stimuli." Additionally, "Given the unique charge reversal design and robust bidirectional deformation mechanism subjected to constant illumination, our work represents an innovative strategy to program nonmonotonous shape transformations and bidirectional locomotion of homogeneous hydrogels using a single constant stimulus for future applications in intelligent actuators and soft robotics."

(2) The complex bidirectional deformation was generated by the hydrogels themselves through a feedback of charge reversal in response to constant external illumination, which does not require any other human control. How to precisely control such charge reversal, specially when the stimulus is not constant?

Our response: We thank this reviewer for the valuable question. In the case of nonconstant illumination, it will be very challenging to precisely control such a charge reversal process as the external stimulus changes.

In our system under constant illumination, the charge reversal originates from the photoisomerization of the two imbedded spiropyran in our hydrogels; therefore, the chemical combination of two spiropyran plays a critical role in this process. With four spiropyran in total, where MCH1 and MCH4 display a much faster speed than MCH2 and MCH3 in response to the same constant illumination, we have four different combinations, namely, MCH(1+2), MCH(1+4), MCH(3+2) and MCH(3+4). Theoretical calculations (Response Figure 3) revealed that these four combinations exhibited significantly different charge reversal curves as a function of irradiation time. Only in the case of MCH(1+4) is the charge reversal finished too quickly due to the fast photoisomerization of both MCH1 and MCH4. This fast charge reversal cannot trigger any macroscale bidirectional deformation of the hydrogel, probably because such a narrow time window prevents sufficient and effective water diffusion, which is a relatively slow process. In addition, the charge reversal behavior could be further controlled by changing the molar ratio of the two spiropyran, which was found to be a useful way to tune the bidirectional bending behaviors of our hydrogels (bending direction symmetry, bending angles). Furthermore, the charge reversal was also affected by the environmental pH, where a higher pH facilitates a relatively fast charge reversal process and is affected by the illumination conditions, where a stronger light intensity generates a relatively fast charge reversal process.

This point has been clarified in the conclusion section on page 23 of the revised manuscript, stating "Specifically, the MCH(1+2) (or MCH(3+4)) hydrogel has to contend with the light penetration depth due to their significant difference in photoisomerization speed, while the MCH(3+2) hydrogel can be considered a more uniform entity because MCH3 and MCH2 possess comparable slow photoisomerization speeds. These two models can generate bidirectional bending deformation when subjected to constant illumination, demonstrating the high robustness and strong universality of such a bidirectional deformation mechanism."; on page 24, stating "through precise control of the charge reversal behavior by selectively mixing two specific spiropyran"

Response Figure 3 for review only. (a) **Figure 2d**. (b) **Figure S17**. (c) **Figure 4d**. (d) **Figure S23**.

(3) *The photoresponsive hydrogels investigated in the paper are structurally homogenous, without any fixed layered/oriented heterogeneous structures, which are dynamic and allow for a highly reversible deformation. In general, compared with heterogeneous actuators, is there any disadvantages of the homogenous hydrogels?*

Our response: The structurally homogenous hydrogels developed here are able to produce a transient gradient in response to light irradiation, driving the generation of the desired deformation. The gradient could be erased in the dark and reproduced upon irradiation, resulting in highly reversible and dynamic actuation. This is the advantage of structurally homogenous hydrogels compared with heterogeneous actuators. A possible disadvantage we could imagine of homogenous hydrogels may be that they cannot be arbitrarily multifunctionalized as heterogeneous actuators. Heterogeneous (e.g., layered) actuators can be assigned different functionalities in different layers.

(4) *The light intensity used in different systems ranged from tens to hundreds of mW/cm². The authors should check whether there is any photothermal effect during the photoactuation experiments.*

Our response: We carried out new experiments to test the possible photothermal effect of different systems using an infrared camera. In the photoactuation experiment, we measured the temperature change in the MCH(1+2) hydrogel before and after irradiation with the strongest photoactuation intensity (450 nm, 26.11 mW/cm²). In the preparation of the SEM samples, a greater light intensity (450 nm, 160.1 mW/cm²) was applied to obtain a more homogenous porosity by improving the light penetration depth, and a fan was used in this experiment to help dissipate possible heat. In these two experiments, we did not observe any obvious increase in the

temperature of the hydrogel sample by thermal imaging. These results showed that no obvious temperature change was observed during this photoactuation process, indicating that no obvious photothermal effect was involved in the photoactuation process. These new results were added to Figure S35 in the revised supporting information, and we also added a sentence on page 23 of the revised manuscript to clarify this point, stating “The driving force for self-regulated bidirectional actuation originates from photoisomerization-induced charge and hydrophilicity variation, and there was no obvious photothermal effect involved (Figure S35).”

A comparable light intensity (450 nm, 154.6 mW/cm²) was also used in the measurement of photoisomerization kinetics using an Ocean Insight DH-2000-BD fiber spectrometer, which is equipped with a temperature controller; therefore, the photothermal effect in this experiment can be ignored.

Figure S35. (a) Measurements of the temperature change in the hydrogel ribbon with different irradiation time (450 nm, 15.24 mW/cm²) in the photoactuation experiment of an **MCH(1+2)** hydrogel strip. (b) Changes in the temperature of the hydrogel dish with irradiation time (450 nm, 160.1 mW/cm²) during preparation of the SEM sample using a dish **MCH(1+2)** hydrogel. Top, photographs of the hydrogel inside the water bath. Bottom, corresponding thermal image of the water bath with the average temperature shown on the top. The scale bar is 10 mm.

(5) What is the real scale of the dual-star-shaped hydrogel shown in Figure 5c? A scale bar is missing and needs to add. The same issue occurred for the hydrogel trips shown in Figure 2b and 4b.

Our response: We apologize for not providing this information in our first version of the manuscript. We have added corresponding scale bars to Figure 5c, Figure 2b and Figure 4b in the revised manuscript.

REVIEWERS' COMMENTS

Reviewer #1 (Remarks to the Author):

The comments have been addressed well. This manuscript is recommended for publishing in Nature Communications.

Reviewer #2 (Remarks to the Author):

The authors have addressed all my concern in full. They also provided additional data for a better elaboration. I recommend for a publication.

Reviewer #3 (Remarks to the Author):

The authors have adequately addressed all the issues raised in the first round review.